# Offline Meta-Reinforcement Learning in Piecewise Stationary Environments

## Abstract

Adapting policies in piecewise stationary environments - where the underlying properties remain stable for periods but abruptly change at unknown points - remains a challenge in reinforcement learning (RL). Addressing this problem using context-based offline meta-RL, which enables generalization to new online tasks from offline data, is particularly appealing, as it avoids the risks associated with online exploration. These methods encode transition history (the context) into a task representation and condition the policy and value function to enable generalization. We show that existing approaches relying on a fixed-length context window face an inherent trade-off between rapid adaptation and inferring a stable task representation in piecewise stationary settings. We overcome this limitation by detecting task changes online from the temporal evolution of task representations and selectively retaining relevant transitions, yielding an adaptive context length. Experiments on continuous control benchmarks demonstrate that our approach enables faster adaptation and stable task identification, resulting in higher-performing policies compared to baselines.

## 1 Introduction

Offline meta-reinforcement learning (OMRL) is the intersection of offline reinforcement learning and meta-learning, in which the goal is to train a generalizable policy from similar datasets, each comprising trajectories for a different task. Prior OMRL methods (Li et al., 2020; 2021; Yuan & Lu, 2022; Gao et al., 2024; Wang et al., 2023; Zhao et al., 2023; Zhang et al., 2025; Nakhaeinezhadfard et al., 2025) use context encoding to extract task-relevant information from trajectories; the context encoder maps previous transitions (the context) to a latent vector, referred to as the task representation, and then conditions the agent on the task representation. Whilst this procedure enables generalization to new, unseen tasks, prior work assumes a stationary environment during test time. That is, the underlying variation factors for different tasks remain the same within an episode.

In many applications, especially real-world scenarios, there could be hidden variation factors within a single episode that change at an arbitrary, unknown timestep, called piecewise stationary. For example, different surfaces have different friction coefficients for a mobile robot, a quadruped robot may encounter diverse terrains, and a resource management system may have to deal with varying incoming workloads (Mao et al., 2019). These hidden variation factors affect the transition dynamics and reward function, and the agent must detect the changes and adapt accordingly for optimal performance. Prior work (Luo et al., 2022; Zhang et al., 2024) limits the number of transitions used to infer the task to $k$-recent ones, enabling adaptation in non-stationary environments. However, the latent task representation can be noisy; therefore, there is a trade-off between rapid adaptation and stable task representation when choosing the hyperparameter $k$. Relying on fewer recent samples (smaller $k$) to infer the task improves adaptation speed, but it yields a less reliable task representation, which adversely affects performance.

Instead of limiting the number of transitions in piecewise stationary environments, we propose to detect the abrupt changes in the environment online and only use relevant transitions to infer the task. Change point detection methods (Gupta et al., 2024) identify abrupt changes in statistical properties of a sequential data stream. These methods have been applied to different fields including healthcare (Gee et al., 2018; Stival

et al., 2023; Johnson & Pedersen, 2025), finance (Kim et al., 2022; Tsaknaki et al., 2025), and cyber-security (Singh & Rathore, 2025). Since the context encoder is trained to map transitions from different tasks – with different variation factors – to distinct latent task representations, the statistical properties of the task representations change in response to the changes in the environment during testing. Building upon this insight, we propose to detect changes in the environment from the sequence of task representations, offering faster adaptation and a more stable task representation.

**Our contributions** are as follows:

**C1** We show that adaptation based on a truncated history of context is sensitive to the context length, where smaller values enable more rapid adaptation at the cost of noisier task inference.

**C2** To overcome this issue, we present ***Change-Aware Meta Learning*** (CAMEL), which detects the changes online from the stream of context using Bayesian online change point detection (BOCPD).

**C3** We show that the naive implementation of BOCPD is prone to false positives due to an imperfect context encoder. To increase robustness to outliers in BOCPD, we adopt and utilize diffusion score matching.

## 2  Background

Our method merges context-based OMRL and change point detection to enable adaptation in piecewise stationary environments. We provide a brief definition of context-based OMRL and change point detection in this section.

### 2.1  Context-based Offline Meta RL

Offline Meta-RL is an extension of offline RL where there are datasets for different, yet similar tasks with different hidden variation factors. Each task is represented as a Markov Decision Process (MDP), $\mathcal{M}_i = \langle \mathcal{S}, \mathcal{A}, R_i, P_i, \gamma, \rho_0 \rangle$, consisting of a shared state space $\mathcal{S}$, action space $\mathcal{A}$, discount factor $\gamma \in [0, 1]$, and initial state distribution $\rho_0(\mathbf{s}_0)$, and task-specific reward function $R_i : \mathcal{S} \times \mathcal{A} \to \mathbb{R}$ and transition dynamics $P_i(\mathbf{s}_{t+1} \mid \mathbf{s}_t, \mathbf{a}_t)$. The objective is to train a meta-policy $\pi$ that can generalize to new tasks, *i.e.*, maximizing the expected cumulative reward over the distribution of test tasks

$$J(\pi) = \mathbb{E}_{\mathcal{M}_i \sim p_{\text{test}}(\mathcal{M}), \, \tau \sim \pi, P_i} \left[ \sum_{t=0}^{T} \gamma^t R_i(\mathbf{s}_t, \mathbf{a}_t) \right], \tag{1}$$

with $\tau = (\mathbf{s}_0, \mathbf{a}_0, \mathbf{s}_1, \dots)$, and $\mathbf{s}_0 \sim \rho_0(\mathbf{s})$, $\mathbf{a}_t \sim \pi(\cdot \mid \mathbf{s}_t)$, $\mathbf{s}_{t+1} \sim P_i(\cdot \mid \mathbf{s}_t, \mathbf{a}_t)$. During meta training, a context encoder $E_\phi : \mathcal{S} \times \mathcal{A} \times \mathbb{R} \times \mathcal{S} \to \mathcal{Z}$ learns a mapping from transitions $\{(\mathbf{s}_j, \mathbf{a}_j, r_j, \mathbf{s}'_j)\}$ to latent a task representation $\mathbf{z}$. Prior work uses different architectures for the context encoder; *e.g.*, FOCAL (Li et al., 2021) uses a multi-layer perceptron (MLP) and then takes the expectation over the latent vectors. In contrast, DORA (Zhang et al., 2024) uses a recurrent neural network (RNN) that models the sequential history of previous transitions. The policy $\pi(\mathbf{s}, \mathbf{z})$ and value functions $Q(\mathbf{s}, \mathbf{a}, \mathbf{z})$ are then conditioned on the task representation $\mathbf{z}$ and trained to maximize Equation (1) using samples from the offline datasets without direct interaction with the environment.

### 2.2  Change Point Detection

Let $\mathbf{Z}_{1:T}$ be a sequence of task representations $\{\mathbf{z}_1, \mathbf{z}_2, \dots, \mathbf{z}_T\}$ where $\mathbf{z}_t \in \mathbb{R}^d$. This sequence $\mathbf{Z}_{1:T}$ is divided into segments $\{\mathbf{Z}_{c_1:c_2-1}, \mathbf{Z}_{c_2:c_3-1}, \dots, \mathbf{Z}_{c_n:c_T}\}$, where $c_i \in \{1, 2, \dots, T\}$ represents $i$-th change point in the environment and $c_1 < c_2 < \cdots < c_n$. Within each segment, we assume that the task representations are sampled from the same underlying distribution. The goal is to identify the change points $(c_i)$ from the sequence of task representations.

Adams & MacKay (2007) and Fearnhead & Liu (2007) illustrated that Bayesian inference for change point detection can be efficient and recursive if the posterior is computed over the most recent change point. In

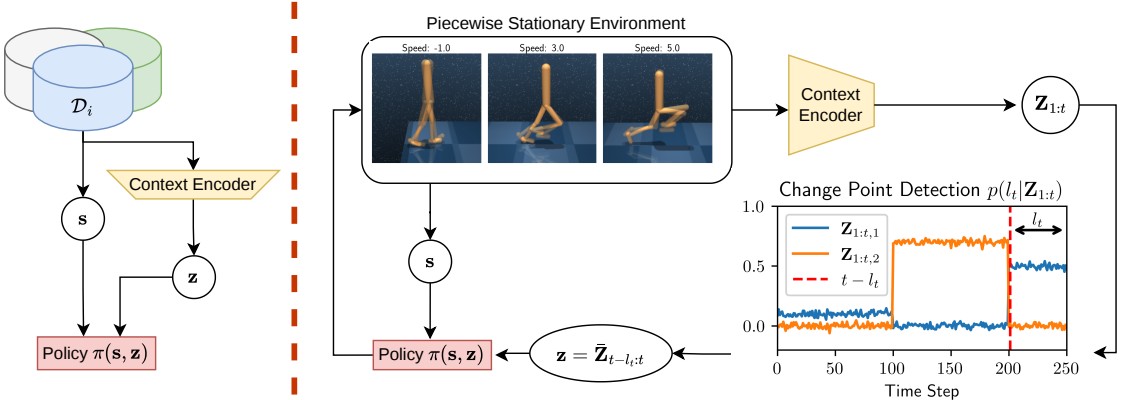

Figure 1: **Method overview. Left:** At the training phase, CAMeL utilizes offline meta-RL to learn distinct task representations **z** for each task, and subsequently the policy is conditioned on the inferred task representation. **Right:** At the testing time, CAMeL detects abrupt changes in the environment from a series of task representations $\mathbf{Z}_{1:t}$ based on change point detection methods. CAMeL then identifies the underlying task based on relevant samples.

other words, the posterior is computed for the length of the current segment $p(l_t|\mathbf{Z}_{1:t}) = \frac{p(l_t, \mathbf{Z}_{1:t})}{\sum_{l_t=0}^{t} p(l_t, \mathbf{Z}_{1:t})}$. The posterior is tractable when the joint distribution between the segment length and task representations is tractable. The joint distribution can be computed using recursion

$$p(l_t, \mathbf{Z}_{1:t}) \propto \sum_{l_{t-1}=0}^{t-1} \underbrace{p(\mathbf{z}_t|\mathbf{Z}_{t-l_t:t-1})}_{\text{Predictive Posterior}} \underbrace{p(l_t|l_{t-1})}_{\text{Prior on change}} p(l_{t-1}, \mathbf{Z}_{1:t-1}). \tag{2}$$

If the predictive posterior in Equation (2) is tractable, then the joint distribution can be computed efficiently. The prior over the current segment length $p(l_t \mid l_{t-1})$ can have a positive probability mass function for $\{0, l_{t-1}+1\}$ since either a change happens or the current segment length increases by one. The assumption is that the task representations in different segments are from the same distribution class $\mathbf{z}_t \sim p(\mathbf{z}_t \mid \theta_i); \theta_i \in \Theta$, but with different parameters for each segment. The predictive posterior in Equation (2) can be written by marginalizing over the parameter space $\theta$

$$p(\mathbf{z}_t|\mathbf{Z}_{t-l_t:t-1}) = \int_{\Theta} p(\mathbf{z}_t|\theta) \underbrace{p(\theta \mid \mathbf{Z}_{t-l_t:t-1})}_{\text{Parameter's Posterior}} d_\theta. \tag{3}$$

According to Bayes' rule, the parameter's posterior in Equation (3) can be derived

$$p(\theta|\mathbf{Z}_{t-l_t:t-1}) \propto p(\theta) \prod_{i=1}^{l_t} p(\mathbf{z}_{t-i} \mid \theta). \tag{4}$$

By assuming different observation models $p(\mathbf{z}|\theta)$, different variants can be derived, *e.g.*, a Gaussian observation model with a conjugate prior on the parameters leads to a tractable predictive posterior, enabling efficient computation of the posterior over the current segment length.

## 3 Method

In this section, we present our method, named CAMeL, which utilizes the OMRL framework to deal with piecewise stationary environments at test time. We assume that the hidden variation factors are fixed for a stochastic period and then abruptly change randomly to another value sampled from the prior distribution.

Since the change is unpredictable, we cannot predict the change points or the hidden variation factors beforehand. Hence, we need to detect the changes in hidden variation factors online. Figure 1 provides a high-level overview of CAMEL.

**Overview**   CAMEL has six main components:

$$\mathbf{z} = \bar{\mathbf{Z}}_{t-l_t:t}, \quad \mathbf{z}_i = E_\phi(\mathbf{s}_i, \mathbf{a}_i, r_i, \mathbf{s}_{i+1}) \tag{5a}$$

Context encoder:

$$\hat{\mathbf{s}}'_t, \hat{r}_t = D_\phi(\mathbf{s}_t, \mathbf{a}_t, \mathbf{z}) \tag{5b}$$

Decoder:

$$q = Q_\psi(\mathbf{s}_t, \mathbf{a}_t, \mathbf{z}) \tag{5c}$$

Q-value function:

$$v = V_\omega(\mathbf{s}_t, \mathbf{z}) \tag{5d}$$

Value function:

$$\mathbf{a}_t \sim \pi_\eta(\mathbf{a}_t \mid \mathbf{s}_t, \mathbf{z}) \tag{5e}$$

Policy:

$$l_t \sim p(l_t | \mathbf{Z}_{1:t}) \tag{5f}$$

Change point detector:

During meta-training, we train the context encoder $E_\phi$, the decoder $D_\phi$, the q-value function $Q_\psi$, the value function $V_\omega$, and the policy $\pi_\eta$ using samples from datasets of different tasks. During testing, the context encoder maps the transitions to the task representation, which is an indirect indicator of the task. The change point detector estimates when the last change occurred from the sequence of task representations $\mathbf{Z}_{1:t}$, as the underlying distribution of the task representation is different after a sudden change in the environment, where $\mathbf{z}_i$ denotes the task representation corresponding to the $i$th transition. Then, only relevant transitions are selected for computing the task representation $\mathbf{z} = \bar{\mathbf{Z}}_{t-l_t:t}$, where $^-$ indicates averaging.

The change point detector assumes that samples in the sequence are independent. However, in practice, transitions are correlated through the policy, whose actions are conditioned on the inferred task representation. During training, the context encoder is optimized to map transitions from different tasks to distinct latent task representations, while transitions are sampled randomly from the replay buffer, largely breaking the temporal correlations, similar to off-policy RL. At deployment, CAMEL relies on the learned context encoder to produce task representations with reduced temporal correlation, thereby making the independence assumption more appropriate for change point detection.

### 3.1   Offline Meta RL

In context-based OMRL methods, the context encoder first identifies the task implicitly by mapping transitions to a task representation $\mathbf{z}$. The meta policy is then conditioned on the task representation and optimized. Most methods (Li et al., 2021; Yuan & Lu, 2022; Gao et al., 2024; Wang et al., 2023; Li et al., 2024; Nakhaeinezhadfard et al., 2025) decouple context encoder training from policy optimization and use separate objectives for task representation learning. In principle, we can use any context-based OMRL method in our framework. For simplicity and effectiveness, we choose a variant of UNICORN (Li et al., 2024) while we show that CAMEL is compatible with other OMRL methods in Sec. C.1.

**Task Representation Learning**   Given a number of transitions from dataset $\mathcal{D}_i$ which correspond to task $\mathcal{M}_i$, the task representation is computed as

$$\mathbf{z}^i = \mathbb{E}_{(\mathbf{s}, \mathbf{a}, r, \mathbf{s}') \sim \mathcal{D}_i}[E_\phi(\mathbf{s}, \mathbf{a}, r, \mathbf{s}')]. \tag{6}$$

UNICORN trains the context encoder to minimize a linear combination of two objectives $\mathcal{L}_{\text{CE}}(\phi) = \mathcal{L}_{\text{Contrastive}}(\phi) + \lambda \mathcal{L}_{\text{Recon}}(\phi)$ where $\mathcal{L}_{\text{Contrastive}}(\phi)$ (Equation (7)) is contrastive loss, $\mathcal{L}_{\text{Recon}}$ (Equation (8)) is the reconstruction loss and $\lambda$ is a hyperparameter for balancing these objectives. We now motivate each of these objectives and explain how to compute them in practice.

Intuitively, transitions from the same task should produce similar task representations, while representations from different tasks should be dissimilar. We use the InfoNCE loss:

$$\mathcal{L}_{\text{Contrastive}}(\theta) = -\sum_i \log \frac{S(\mathbf{z}^i, \bar{\mathbf{z}}^i)}{\sum_j S(\mathbf{z}^i, \bar{\mathbf{z}}^j)}, \tag{7}$$

where $\bar{\mathbf{z}}^i = \lambda \mathbf{z}^i + (1 - \lambda)\bar{\mathbf{z}}^i$ is the moving average of task representations controlled by $\lambda$, and $S(\mathbf{z}^i, \mathbf{z}^j) = \exp(-\|\mathbf{z}^i - \mathbf{z}^j\|_2^2/\alpha)$ measures similarity. Positive samples are obtained from the moving average of the same task representation, while negatives are drawn from other tasks. The moving average stabilizes training by smoothing updates. This objective provides a lower bound on the mutual information between tasks and task representations, $I(\mathbf{z}; \mathcal{M})$ (Zhang et al., 2024).

Due to limited coverage of datasets, the context encoder may ignore task-relevant information (reward function and dynamics) and focus on task-agnostic information (state and action) to identify the task, reducing generalization. UNICORN utilized a decoder to predict dynamics and reward conditioned on the task representation and then trained the decoder jointly with the context encoder. The reconstruction objective reduces this context distribution shift and enables the context encoder to focus on task-relevant information

$$\mathcal{L}_{\text{Recon}}(\phi) = \|\hat{\mathbf{s}}' - \mathbf{s}'\|_2^2 + (\hat{r} - r)^2 \tag{8}$$

where $\hat{\mathbf{s}}'$ is the predicted next state and $\hat{r}$ is the predicted reward according to Equation (5b). In Equation (8), we normalize the reward and the next state to have zero mean and unit variance due to different magnitudes between next states and rewards.

**Meta Policy Optimization**    Context-based OMRL is an extension of offline RL where the policy and value functions are conditioned on the task representation. A well-known challenge in offline RL is distribution shift, which arises from out-of-distribution (OOD) action selection during temporal difference (TD) learning (Levine et al., 2020). Actor-critic methods without regularization overestimate the value function while the policy is trained to optimize the value function. We adopt *Implicit Q-Learning* (IQL Kostrikov et al., 2022) to sidestep this issue. IQL utilizes expectile regression in policy evaluation to predict the upper expectile of the TD targets in SARSA style without querying OOD actions. The value function then approximates an expectile with respect to only the action distribution

$$L_V(\omega) = \mathbb{L}_2^\tau(Q_{\bar{\psi}}(\mathbf{s}, \mathbf{a}, \mathbf{z}) - V_\omega(\mathbf{s}, \mathbf{z}))$$

where $\mathbb{L}_2^\tau(x) = (\tau - \mathbb{1}(x < 0))x^2$ is $\tau$ expectile regression and $\bar{\psi}$ is exponential moving average of $\psi$. The value function is then used in computing the target for training the Q-value function

$$L_Q(\psi) = \|r + \gamma V_\omega(\mathbf{s}', \mathbf{z}) - Q_\psi(\mathbf{s}, \mathbf{a}, \mathbf{z})\|_2^2.$$

The policy is optimized based on advantage-weighted regression (Peng et al., 2019)

$$L_\pi(\eta) = -\log \pi_\eta(\mathbf{a} \mid \mathbf{s}, \mathbf{z}) \exp\left(\mathcal{B}A(\mathbf{s}, \mathbf{a}, \mathbf{z})\right)$$

where $A(\mathbf{s}, \mathbf{a}, \mathbf{z}) = Q_\psi(\mathbf{s}, \mathbf{a}, \mathbf{z}) - V_\omega(\mathbf{s}, \mathbf{z})$ is the advantage function and $\mathcal{B} \in [0, \infty)$ is the inverse temperature.

### 3.2   Robust Change Point Detection

In Bayesian online change point detection (BOCPD), if the observation model is a poor description of the sequence of task representations, uncertainties are not reliable, and posterior inferences are sensitive to outliers (Matsubara et al., 2022). To improve robustness to outliers, we use diffusion score matching in generalized Bayesian inference as proposed by Altamirano et al. (2023).

**Generalized Bayesian Inference**    Given a sequence of task representations $\mathbf{Z}_{i:j}$, the generalized posterior over the observation model's parameters is defined as

$$p(\theta|\mathbf{Z}_{i:j}) \propto p(\theta) \exp\left(-\alpha T \mathcal{L}(\theta)\right), \tag{9}$$

where $\alpha > 0$ is a scaling parameter (aka, the learning rate) that determines how quickly the posterior learns from the data, and $L(\theta)$ can represent any loss function over the parameter space $\Theta$. The standard posterior update can be derived by setting $\alpha = 1$ and using the average negative log-likelihood $\mathcal{L}(\theta) = \frac{1}{T} \sum_{t=i}^{j} -\log p(\mathbf{z}_t \mid \theta)$ as the loss function. Decreasing the value of $\alpha$ increases the robustness of change point detection by making the detector more conservative. Unless otherwise specified, CAMEL uses $\alpha = 0.1$ as the default value.

---

**Algorithm 1** Robust change point detection

---

**Input**: Sequence of task representations $\mathbf{Z}_{1:T}$, prior $p(l_t|l_{t-1})$, scaling parameter $\alpha$.

1: Initialize $l_0 = 0$ and $p(l_0, \mathbf{z}_1) = 1$.
2: **for** $t = 2$ to T **do**
3:     **for** $t' = 1$ to $t - 1$ **do**
4:         Compute the posterior over model parameters $p(\theta \mid \mathbf{Z}_{t-t':t-1})$ according to Equation (13).
5:         Compute predictive posterior $p(\mathbf{z}_t|\mathbf{Z}_{t-t':t-1})$ according to Equation (3).
6:         Compute joint distribution $p(l_t = t', \mathbf{Z}_{1:t})$ according to Equation (2).
7:     **end for**
8:     Set $l_t = \arg\max_{t'} p(l_t = t', \mathbf{Z}_{1:t})$.
9: **end for**
10: **Output** $l_{1:T}$

---

**Diffusion Score Matching** In diffusion score matching (Barp et al., 2019), the loss function $\mathcal{L}(\theta)$ in Equation (9) utilizes a weighted version of the Fisher divergence between the statistical model $p_\theta$ and the data-generating process $p_0$

$$L(\theta) = \|m^T(\mathbf{z})\big(s_{p_\theta}(\mathbf{z})) - s_{p_0}(\mathbf{z}))\big)\|_2^2, \tag{10}$$

where $m$ is a point-wise invertible matrix function $m : \mathbf{z} \in \mathbb{R}^d \to \mathbb{R}^{d \times d}$, also known as the diffusion matrix (Anastasiou et al., 2023), and $s_p(\mathbf{z})$ is the score function of density $p$ defined as $s_p(\mathbf{z}) = \nabla \log p(\mathbf{z})$. Estimating $\mathcal{L}(\theta)$ in Equation (10) directly is challenging, as it would require estimating the unknown score of the true generating process $s_{p_0}$. Liu et al. (2022) showed that Equation (10) can be approximated under smoothness and boundary conditions as

$$L(\theta) \approx \|m(\mathbf{z})^T s_{p_\theta}(\mathbf{z})\|_2^2 + 2\nabla.\big(m(\mathbf{z})m(\mathbf{z})^T s_{p_\theta}(\mathbf{z})\big). \tag{11}$$

Equation (11) does not depend on the data-generating process $p_0$ and can be computed from the sequence of task representations. The posterior in Equation (9) has a closed-form solution for exponential family observation models in the form of $p_\theta(\mathbf{z}) = \exp\big(\eta(\theta)^T r(\mathbf{z}) - a(\theta) + b(\mathbf{z})\big)$ with conjugate priors (Altamirano et al., 2023)

$$p(\theta|\mathbf{Z}_{i:j}) \propto p(\theta) \exp\big(-\alpha T(\eta(\theta)^T \Gamma \eta(\theta) + \eta(\theta)^T \nu), \tag{12}$$

where $\Gamma = \frac{1}{T}\sum_{t=i}^{j}(\nabla r^T mm^T \nabla r)(\mathbf{z}_t)$ and $\nu = \frac{2}{T}\sum_{t=i}^{j}\nabla(r^T mm^T \nabla b + \nabla.(mm^T \nabla r))(\mathbf{z}_t)$. Considering a Gaussian observation model ($z_t \in \mathbb{R}^1$) where

$$\eta(\theta) = \theta = [\frac{\mu}{\sigma^2}, \frac{1}{\sigma^2}]^T, \quad r(z) = [z, -\frac{1}{2}z^2]^T, \quad a(\theta) = \frac{\mu^2}{2\sigma^2} + 2\log\sigma, \quad b(z) = \log\frac{1}{\sqrt{2\pi}}$$

with a conjugate prior in the form of $p(\theta) \propto \exp\big(-\frac{1}{2}(\theta-\mu_0)^T \Sigma_0^{-1}(\theta-\mu_0)\big)$ and choosing the diffusion matrix to be $m(z) = \frac{1}{\sqrt{1+z^2}}$ leads to a closed-form solution for the posterior $p(\theta \mid \mathbf{Z}_{i:j}) \propto \exp\big(-\frac{1}{2}(\theta-\mu_T)^T \Sigma_T^{-1}(\theta-\mu_T)\big)$ where

$$\Sigma_T^{-1} = \Sigma_0^{-1} + 2\alpha \sum_{t=i}^{j} \frac{1}{1+z_t^2}\begin{bmatrix} 1 & -z_t \\ -z_t & z_t^2 \end{bmatrix}, \quad \mu_T = \Sigma_T\big(\Sigma_0^{-1}\mu + 2\alpha \sum_{t=i}^{j} \frac{1}{(1+z_t^2)^2}\big)\begin{bmatrix} 2z_t \\ 1-z_t^2 \end{bmatrix}. \tag{13}$$

This formulation can be extended by considering a multivariate Gaussian observation model with a diagonal diffusion matrix. The goal is to compute the posterior over the current segment length $p(l_t \mid \mathbf{Z}_{1:t})$. We summarize the procedure for robust change point detection in Alg. 1. The difference between robust BOCPD and BOCPD is in computing $p(\theta \mid \mathbf{Z}_{t-l_t:t-1})$.

**Computation Complexity** A naive implementation of BOCPD will compute the posterior over all current segment lengths, leading to the computation complexity of $\mathcal{O}(\sum_{t=1}^{T}(t)) = \mathcal{O}(T^2)$. In the online version, the computation cost at each step increases linearly, which can be limiting for real-time control on a long horizon. To mitigate this issue, only the $k$ most probable current segment lengths are stored for computing the posterior in the following steps, reducing the overall computation complexity to $\mathcal{O}(kT\log T)$. We compare the computation cost of CAMEL to baselines with truncated history in Figure 7 (Sec. C.2).

### 3.3 Impact of Diffusion Score Matching on Posterior (Illustrative Example)

In this section, we provide an example to justify using diffusion score matching in CAMEL. In BOCPD, if the observation model is a poor description of the sequence of task representations, uncertainties are not reliable; consequently, posterior inferences are sensitive to outliers.

**Assumption 3.1.** Let $z_t \in \mathbb{R}^1$ be a sequence of task representations corresponding to one task (no change in the environment), then

$$z_t \sim (1 - \epsilon)\mathcal{N}(\mu_1, \sigma_0^2) + \epsilon\mathcal{N}(\mu', \sigma_0^2),$$

where $\mu_1$ is the mean of the task representation corresponding to the underlying task.

This assumption indicates that with probability of $\epsilon$, the context encoder misidentifies the corresponding task. Figure 8 in Sec. C.3 provides empirical evidence for this assumption by illustrating the task representations for different tasks. When considering a Gaussian observation model $p(z_t \mid \theta) = \mathcal{N}(\cdot \mid \mu, \sigma)$, we want to compute the parameters $\mu$ and $\sigma$ for Bayesian inference (*Gaussian*),

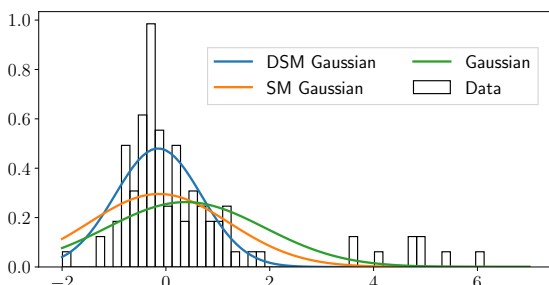

Figure 2: The impact of outliers in the posterior. **Diffusion score matching increases the robustness to outliers**.

generalized Bayesian inference with score matching $m(z) = 1$ (*SM Gaussian*), and generalized Bayesian inference with diffusion score matching *DSM Gaussian*. We conduct a numerical simulation where $\mu_1 = 0$, $\mu' = 0.75$, $\sigma_0 = 0.75$, and $\epsilon = 0.1$. Figure 2 illustrate the results for $n = 100$ samples with $\alpha = 0.1$. While score matching slightly increased robustness to outliers relative to Bayesian inference, diffusion score matching is the most robust variant, effectively ignoring outliers in the posterior.

## 4 Experiments

In this section, we evaluate CAMEL in a set of non-stationary MuJoCo (Todorov et al., 2012) and Contextual DeepMind Control Suite (Tunyasuvunakool et al., 2020; Rezaei-Shoshtari et al., 2023) environments. We provide more details in Sec. B and more experiments in Sec. C. Our experiments seek to answer the following questions:

**Q1** Does CAMEL's adaptive context length based upon change point detection enable rapid and accurate adaptation in piecewise stationary environments? In particular, is it more effective than prior methods based on fixed-length truncated histories?

**Q2** How does diffusion score matching impact the accuracy and the delay in change point detection, and how much of an impact does it have on downstream RL performance?

**Baselines** To evaluate the adaptation performance of our method, we compare it to the following baselines:

- **UNICORN** (Li et al., 2024) is an OMRL method that uses all of the transitions to infer the task representation. This baseline illustrates the necessity of inferring the task representation from only relevant transitions. For this method, we use the same hyperparameters as our OMRL backbone.

- **ESCP** (Luo et al., 2022) is an online context-based meta-RL method with an RNN context encoder. To enable adaptation to non-stationary environments, the context length (transition length) is limited to the $k$ most recent when computing the task representation $z$. The context encoder is trained to minimize variance and maximize the determinant of the relational matrix.

- **DORA** (Zhang et al., 2024) is the closest baseline to CAMEL. It uses offline datasets with different variation factors to enable adaptation in non-stationary testing. Similarly to ESCP, $k$ recent transitions are used to compute the task representation **z** while we detect the change points instead.

Table 1: **Improved adaptation with change point detection in piecewise stationary testing**. Average normalized return over 15 experiments, $\pm$ represents standard deviation. The last three rows indicate variants of CAMeL. **Bold** indicates the highest value with statistical significance according to the t-test with p-value $< 0.05$.

| Agent | Hopper Friction | Walker Friction | Ant Direction | Cheetah Speed | Cheetah L/S | Finger L/S |
|---|---|---|---|---|---|---|
| UNICORN $k = \infty$ | $81.49 \pm 4.39$ | $78.84 \pm 4.32$ | $58.65 \pm 6.97$ | $6.12 \pm 11.78$ | $64.55 \pm 9.12$ | $82.69 \pm 6.86$ |
| UNICORN $k = 8$ | $92.60 \pm 4.22$ | $93.93 \pm 3.81$ | $86.38 \pm 4.96$ | $85.69 \pm 10.94$ | $94.97 \pm 2.57$ | $90.36 \pm 6.23$ |
| UNICORN $k = 4$ | $92.81 \pm 4.10$ | $94.69 \pm 2.62$ | $92.76 \pm 7.17$ | $89.17 \pm 13.74$ | $94.57 \pm 3.74$ | $96.36 \pm 2.84$ |
| UNICORN $k = 2$ | $89.76 \pm 7.65$ | $94.58 \pm 3.09$ | $86.27 \pm 7.55$ | $81.64 \pm 12.20$ | $94.97 \pm 3.15$ | $93.49 \pm 5.46$ |
| ESCP $k = 8$ | $86.64 \pm 5.56$ | $59.07 \pm 6.31$ | $88.54 \pm 8.43$ | $65.09 \pm 12.23$ | $65.15 \pm 0.81$ | $73.60 \pm 6.36$ |
| ESCP $k = 4$ | $86.60 \pm 5.37$ | $67.84 \pm 9.42$ | $83.91 \pm 10.58$ | $75.64 \pm 11.94$ | $67.97 \pm 3.09$ | $78.49 \pm 2.43$ |
| ESCP $k = 2$ | $84.16 \pm 12.21$ | $61.49 \pm 6.96$ | $68.90 \pm 5.94$ | $73.42 \pm 6.94$ | $66.95 \pm 3.27$ | $83.49 \pm 5.76$ |
| DORA $k = 8$ | $93.09 \pm 5.85$ | $82.10 \pm 5.21$ | $80.66 \pm 6.84$ | $62.81 \pm 10.55$ | $75.10 \pm 11.48$ | $87.63 \pm 5.35$ |
| DORA $k = 4$ | $96.66 \pm 3.45$ | $85.13 \pm 4.73$ | $75.40 \pm 11.60$ | $60.79 \pm 11.50$ | $76.07 \pm 15.62$ | $89.89 \pm 5.53$ |
| DORA $k = 2$ | $93.15 \pm 2.99$ | $85.28 \pm 7.10$ | $57.34 \pm 10.76$ | $57.66 \pm 10.01$ | $72.57 \pm 10.32$ | $88.54 \pm 5.70$ |
| MetaDT | $96.15 \pm 0.99$ | $84.5 \pm 10.06$ | $88.30 \pm 5.24$ | $87.28 \pm 2.00$ | $87.78 \pm 1.24$ | $95.84 \pm 1.61$ |
| Gaussian (ours) | $96.37 \pm 6.32$ | $96.40 \pm 3.29$ | $96.18 \pm 3.36$ | $97.67 \pm 1.05$ | $98.21 \pm 1.38$ | $98.65 \pm 1.51$ |
| SM Gaussian (ours) | $97.53 \pm 3.04$ | $98.72 \pm 2.27$ | $96.36 \pm 2.91$ | $98.39 \pm 1.52$ | $98.72 \pm 1.21$ | $98.53 \pm 0.81$ |
| DSM Gaussian (ours) | $\mathbf{99.72} \pm 0.27$ | $99.88 \pm 2.91$ | $\mathbf{99.21} \pm 1.06$ | $99.36 \pm 1.12$ | $\mathbf{99.76} \pm 0.35$ | $\mathbf{99.67} \pm 0.71$ |

- **MetaDT** (Wang et al., 2024) frames offline meta-RL as conditional sequence modeling and utilizes a transformer architecture to predict actions conditioned on a sequence of previous states, actions, rewards, desired return, and task representation. The task representation is obtained from an RNN context encoder trained via reconstruction.

To ensure a fair comparison, we use the same offline RL (IQL) with the same hyperparameters across all OMRL baselines.

**Offline Datasets** We use the Dropout Q-function (DroQ, Hiraoka et al., 2022) to generate the datasets, and we train a separate agent for each dataset. The dataset consists of trajectories collected from rolling out the corresponding DroQ agent at different training stages; each dataset contains 1000 trajectories. We collect 20 datasets for each environment with different variation factors.

### 4.1 CAMeL's Adaption Performance

We evaluate the performance of CAMeL alongside baselines in piecewise stationary environments. We consider the following variants of BOCPD in CAMeL:

- *Gaussian* where we assume a multivariate Gaussian observation model and we update the predictive posteriors based on Bayesian inference.

- *SM Gaussian* where the predictive posteriors are updated based on generalized Bayesian inference with score matching instead of Bayesian inference.

- *DSM Gaussian* where diffusion score matching is utilized in general Bayesian inference.

For each environment, we evaluate three distinct scenarios, each defined by different values of the variation factors. In every scenario, the variation factors change abruptly every 100 time steps, with each trial lasting 500 time steps in total. Importantly, CAMeL makes no assumptions about the segment lengths within each trial. CAMeL assumes a uniform prior over the segment length, assigning equal probability mass to segment lengths, *e.g.*, $p(l_t \mid l_{t-1}) = \frac{1}{h}$, where the default horizon is set to $h = 100$. We repeat each scenario with 5 random seeds, resulting in 15 experiments per environment. Results are summarized in Table 1, where returns are normalized such that a random policy scores 0 and an expert OMRL agent with access to true change points scores 100. In some experiments, the agent slightly outperforms the expert OMRL agent, *e.g.* in the *Walker Friction* environment. Although the policy relies on implicitly identifying the underlying task through its task representation, the inferred task representation and policy do not always perfectly align with the true underlying task, which can occasionally lead to better performance than the OMRL agent

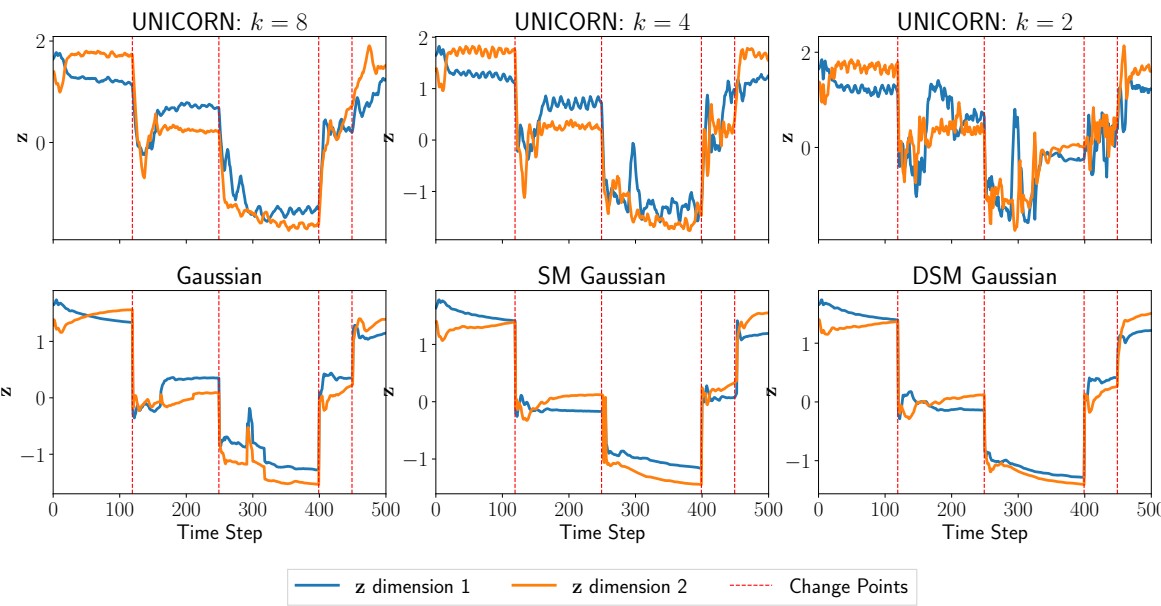

Figure 3: **Visualizing the task representation**: Fewer transitions in the truncated history increase adaptation speed at the cost of a noisier task representation. We sidestep this issue by detecting change points in CAMEL.

given true change points. Moreover, the inferred task representations of DSM Gaussian and the OMRL agent with access to the true change points can be nearly identical when change points are detected accurately. The resulting small differences in task representations, combined with the imperfect conditional policy and the inherent stochasticity of neural network optimization, can occasionally lead to slightly higher returns.

In most environments, UNICORN with all context performs worst, illustrating that in piecewise stationary environments, it is crucial to infer the task from relevant samples. ESCP and DORA with RNN context encoders underperform UNICORN for different values of $k$ in most of the environments, indicating that the task representation learning in UNICORN is superior.

Decreasing the value of $k$ from 8 to 4 improves the performance of ESCP, DORA, and UNICORN in most environments; however, when we further reduce $k$ to 2, there is a decrease in performance. We hypothesize that larger values of $k$ reduce adaptation speed, but lead to more stable task representation when the environment is stationary, since the task representations can be noisy. We provide further evidence in Sec. 4.2.

Different variations of CAMEL outperform baselines as illustrated in Table 1. However, the *SM Gaussian* and *DSM Gaussian* variants are more performant than the *Gaussian variant*. This suggests that diffusion score matching improves robustness to outliers when detecting the change points, increasing the reliability of task representation in testing. We provide evidence in Sec. 4.3.

## 4.2 Task Representation Visualization

In this section, we investigate how truncated history and change point detection strategies affect computing the task representation from transitions in piecewise stationary testing. The context length $k$ in truncated history affects the performance in testing, as illustrated in Table 1. We hypothesize that using fewer recent transitions improves the adaptation speed, but results in a noisier task representation. To evaluate this hypothesis, we visualize the task representation learning for different values of the number of recent transitions $k$ and different variants of BOPCD used in CAMEL in Figure 3 for the *Cheetah Length Speed* environment. We use the same context encoder and policy for different experiments. Note that, unlike Table 1, the abrupt changes occur at irregular intervals. When using $k = 8$ recent samples, the task representation is more stable within each segment compared to $k = 4$. Still, after an abrupt change occurs

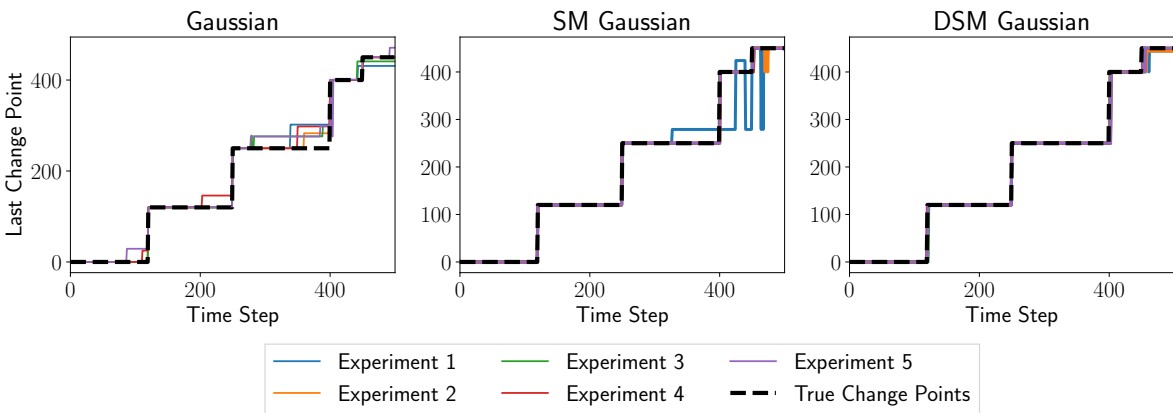

Figure 4: **Comparison of different variants of BOCPD**: BOPCD is prone to false positives. General Bayesian inference with (diffusion) score matching increases the robustness to outliers.

(vertical dashed red lines), it takes more time steps to adapt the representation. We observe the same pattern when decreasing the number of recent transitions to $k = 2$.

The task representation within each segment is more stable when using the change point detection strategy compared to the truncated history, and the task representation adapts quickly after detecting the change. However, BOCPD with the Gaussian model is prone to (temporary) false positives. We sidestep this issue by improving the robustness of BOCPD with diffusion score matching. Note that the sequence of task representations $\mathbf{Z}_{1:T}$ is different at each experiment. At each time step, computing the task representation $z$ from the sequence $\mathbf{Z}_{1:t}$ affects future transitions and subsequently future task representations. Inferring a different task leads to selecting actions that are (sub-)optimal for the wrong task. This can increase the distribution shift between collected transitions and offline datasets, reducing the reliability of task inference. This cycle of inaccurately identifying the task and increasing the context distribution shift can impair performance.

Sec. C.3 visualizes the evolution of task representation $\mathbf{z}$ for other environments.

### 4.3 Change Detection in Different Variants of CAMeL

We next isolate the effect of Generalized Bayesian Inference with diffusion score matching on change point detection in CAMEL. As illustrated in Table 1, using diffusion score matching in generalized Bayesian inference can improve the performance Detecting the wrong change points hinders the performance; false negatives (delays or even not detecting the change) reduce the adaptation speed in task representation, and false positives lead to noisier task representation due to fewer transitions. Figure 4 compares the detected change points by different variants to the true changes in the environment for *Finger Length Speed*.

The number of false negatives (delays) in the predictions of BOCPD with the Gaussian model is lower. However, the predictions contain false positives, reducing the stability in the task representation. We sidestep this issue by using (diffusion) score matching in general Bayesian inference when computing the posterior over the parameters of the observation model $p(\theta \mid \mathbf{Z}_{1:t})$. While more robust variants reduce the sensitivity to outliers, it slightly increases delays in detecting the change points, as illustrated in Table 2.

Table 2: Total delays and false positives for detecting change points. **Diffusion score matching results in a slight increase in detection delay, but reduces the rate of false positives.** Averaged over 5 random seeds where $\pm$ indicates standard deviation.

| Variant | Gaussian | SM Gaussian | DSM Gaussian |
|---|---|---|---|
| Delay | $1.2 \pm 1.6$ | $8.0 \pm 14.5$ | $3.2 \pm 1.3$ |
| False Positive | $3.2 \pm 1.3$ | $1.2 \pm 1.6$ | $0.2 \pm 0.4$ |

However, this increase in detection delay is insignificant (less than one time-step), and the robust variant of CAMeL performs almost identically to the OMRL agent when given true change points.

The *SM Gaussian* variant achieves a detection delay of $8.0 \pm 14.5$ (Figure 4), indicating a larger mean and substantially higher standard deviation than the *Gaussian* variant. As shown in Figure 4, this is primarily caused by Experiment 1 (blue line), where the method fails to detect the change point in the fourth interval (between 400 and 450 time steps). This missed detection acts as an outlier, substantially increasing both the mean and standard deviation of the detection delay. Excluding this outlier reduces the detection delay to $2.0 \pm 0.83$.

### 4.4 Further Results

Sec. C provides additional experiments and results. Sec. C.1 shows that CAMeL is also compatible with other offline meta-RL methods. Sec. C.2 compares the computational cost of CAMeL with that of the baselines. Although change point detection introduces additional computational overhead, CAMeL remains suitable for real-time control, requiring only approximately $0.0035\,\mathrm{s}$ per control step. Sec. C.3 visualizes the evolution of task representations, illustrating that CAMeL enables rapid adaptation with stable task inference. Sec. C.5 investigates the performance of CAMeL with irregular change intervals, showing that it can detect the changes and outperforms the truncated history baselines. Sec. C.6 compares the performance for different change intervals, showing that CAMeL outperforms truncated history regardless of the frequency of abrupt changes. We provide ablation studies on the impact of scaling parameter $\alpha$ (Sec. C.7) and task representation dimension (Sec. C.8) on the performance. CAMeL is performant for a wide range of scaling parameters $\alpha$, and the default value of $\alpha = 0.1$ is suitable for different environments, while higher values increase the sensitivity in change point detection. Task representation $\mathbf{z}$ with higher dimensions slightly improves performance due to better task distinguishability at the cost of more computation. We also evaluate CAMeL with CUSUM change point detection in Sec. C.9, which is sensitive to the choice of threshold, requiring careful tuning in different experiments. Sec. C.10 presents an ablation study on the segment-length prior (horizon $h$), demonstrating that CAMeL is robust to this hyperparameter. In addition, Sec. C.11 investigates the effect of the pruning parameter $k$ on both performance and computational cost. Sec. C.12 investigates the performance of MACAW (Mitchell et al., 2021) and BOReL (Dorfman et al., 2021) in our settings.

## 5 Related Work

We provide a brief overview of piecewise stationary RL and change point detection, as CAMeL enables adaptation based on change point detection.

### 5.1 Piecewise Stationary Reinforcement Learning

In piecewise stationary environments, transition dynamics and reward functions can change abruptly within each episode. Optimal performance requires policies that adapt online in response to these changes. Alegre et al. (2021) employs model-based RL with an ensemble of transition dynamics networks, where each network models the dynamics for a specific set of variation factors. However, this approach is limited in that a new network must be trained from scratch for each new set of variation factors. MOLe (Nagabandi et al., 2019) meta-trains a prior dynamics model for model-based RL using Model-Agnostic Meta-Learning (MAML; Finn et al. 2017) while maintaining a growing mixture of task-specific dynamics models. During non-stationary execution, the most suitable model is selected at each step, and its parameters are adapted online using the EM algorithm. Although MOLe is more flexible than CAMeL, it incurs substantially higher computational cost. Furthermore, unlike CAMeL, which is trained entirely from offline datasets, MOLe requires online interactions for adaptation.

A more flexible approach leverages multi-modal distributions, *e.g.*, Gaussian mixture models (GMMs), for the task representation (Bing et al., 2023; Qi et al., 2025) in the meta-RL setting. However, selecting appropriate hyperparameters (*e.g.*, the number of components) remains challenging, and GMMs are prone to mode collapse (Soletskyi et al., 2025).

SeCBAD (Chen et al., 2022) is the most closely related work to CAMEL, as both methods jointly infer the segment length and the task representation (referred to as the belief in SeCBAD) from trajectories. SeCBAD trains a GRU-based context encoder and reconstruction decoders, and iteratively estimates the posterior over segment lengths from reconstruction likelihoods. In contrast, CAMEL leverages offline meta-RL with contrastive representation learning to learn task representations that distinctly separate different tasks, and infers segment lengths from the temporal evolution of these task representations.

Alternative methods (Luo et al., 2022; Zhang et al., 2024) use only the $k$ most recent transitions for task inference, trading off inference stability for faster adaptation to non-stationary dynamics. In contrast, CAMEL does not require a multi-modal distribution for latent task representation. Rather than using a fixed-length context, we identify changes online and dynamically select relevant transitions for the context. This adaptive-length context enables both stable task inference and rapid adaptation to distribution shifts.

### 5.2 Change Point Detection

Change-point detection aims to identify time steps at which the statistical properties of a time series undergo abrupt changes. Offline methods (Maillard, 2019a) can localize change points accurately by leveraging the entire dataset, but their non-causal nature renders them unsuitable for real-time applications. Sequential frequency-based methods (Lai & Xing, 2010; Maillard, 2019b; Wang & Ning, 2025), such as Cumulative Sum (CUSUM; Page, 1954) and the Generalized Likelihood Ratio (GLR; Capizzi, 2001), monitor summary statistics—typically cumulative deviations from a reference distribution—and signal a change when a predefined threshold is exceeded. However, these approaches are sensitive to parametric assumptions and threshold selection (Romano et al., 2023). Bayesian Online Change Point Detection (BOCPD; Fearnhead & Liu, 2007) addresses these limitations by maintaining a posterior distribution over the current segment length (run length), enabling principled uncertainty quantification over both the occurrence and timing of change points. Despite this advantage, BOCPD remains sensitive to model misspecification and outliers. To improve robustness, Knoblauch & Damoulas (2018) introduced generalized Bayesian inference based on $\beta$-divergence, though this approach incurs substantial computational cost as it requires multiple variational approximations at each time step. More recently, Altamirano et al. (2023) proposed a diffusion score-matching formulation of generalized Bayesian inference, yielding closed-form posterior updates for exponential family models. We adopt this formulation for detecting abrupt changes in CAMEL due to robustness and computation efficiency.

## 6 Conclusion

This paper investigates offline meta-RL for piecewise stationary environments. First, we show that existing methods relying on a fixed window of context history are suboptimal, as there is an inherent trade-off between stable task inference and rapid adaptation. To overcome this issue, we introduce Change-Aware Meta Learning (CAMEL), which detects changes online from the temporal evolution of task representation. We utilize a Bayesian framework for change point detection while leveraging diffusion score matching in generalized Bayesian inference to increase robustness to outliers. Experiments show that our approach yields rapid adaptation when a change occurs while maintaining consistent behavior between change points.

**Limitations** We assume that the changes in the environment are sudden and unpredictable. While this assumption can capture a wide range of applications, there are some exceptions. The variation factors can change slowly over time, *e.g.* variation in the temperature affecting the hydraulic dynamics (Egli & Hutter, 2022). Also, there might be hidden patterns in the changes in the environments where recognizing these patterns beforehand is necessary for optimal performance. Furthermore, CAMEL's performance depends on the quality of the context encoder used for task inference and change point detection, which in turn is influenced by the quality, diversity, and size of the training dataset.

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

Table 3: Table of Notations.

| Notation | Description |
| --- | --- |
| $A$ | Advantage function |
| $\mathcal{A}$ | Action space |
| $\mathbf{a}_t$ | Action at time step $t$ |
| $\alpha$ | Scaling parameter in generalized Bayesian inference |
| $\mathcal{B}$ | Inverse temperature in advantage weighted regression |
| $\beta$ | Weight of inverse distance in contrastive objective |
| $c$ | Change point |
| $\mathcal{D}_i$ | Dataset of trajectories for task $i$ |
| $D_\phi$ | Decoder |
| $E_\phi$ | Context encoder |
| $l_t$ | Segment length at time step $t$ |
| $\mathcal{M}_i$ | Task $i$ modeled as MDP |
| $m$ | Diffusion matrix |
| $P_i$ | Transition dynamics for task $i$ |
| $Q_\psi$ | Q-value function |
| $R_i$ | Reward function for task $i$ |
| $r_t$ | Reward at time step $t$ |
| $\hat{r}$ | Predicted reward |
| $\mathcal{S}$ | State space |
| $\mathbf{s}_t$ | State at time step $t$ |
| $\mathbf{s}'_t$ | Next state at time step $t$ |
| $\hat{s}'$ | Predicted next state |
| $s_{p_\theta}$ | Score function of the distribution $p_\theta$ |
| $V_w$ | Value function |
| $\mathbf{x}_t$ | Observation at time step $t$ |
| $\mathbf{z}$ | Task representation |
| $\mathbf{Z}$ | Sequence of task representations |
| $\gamma$ | Discount factor |
| $\lambda$ | Weight of reconstruction objective |
| $\nabla.$ | Divergence operator |
| $\pi_\eta$ | Policy |
| $\rho_0$ | Initial state distribution |
| $\tau$ | Expectile of value function |
| $\theta$ | Parameter of observation model |

# A Extended Related Work

CAMEL enables real-time adaptation to piecewise stationary environments based on the (offline) meta-RL framework. In this section, we aim to provide a brief overview of meta-RL and adaptation in RL.

## A.1 Meta Reinforcement Learning

While RL can train expert policies for complex tasks, these policies often fail to generalize even to small perturbations. Meta-RL addresses this limitation by equipping policies with the ability to adapt to new, similar tasks. Meta-RL methods train on a distribution of related tasks, learning a meta-policy that maximizes expected cumulative rewards across the task distribution. At test time, the learned meta-policy can accomplish new tasks from the same distribution within a few trials. Two main paradigms exist: gradient-based and context-based meta-RL. Gradient-based approaches (Finn et al., 2017; Rothfuss et al., 2019; Tang et al., 2021) update the meta-policy parameters during testing, while context-based approaches (Lee et al., 2020; Zintgraf et al., 2021; Choshen & Tamar, 2023; Beck et al., 2023; 2024; Wen et al., 2024) enable adaptation without parameter updates by conditioning the policy on samples from the new task, referred to as the *context*. Offline meta-RL (OMRL) extends this framework to learn generalizable policies from offline datasets. Most OMRL methods adopt the context-based paradigm (?Li et al., 2021; Gao et al., 2024; Zhao et al., 2023; Wang et al., 2023; Zhou et al., 2024; Xu et al., 2022; Wang et al., 2024; Ni et al., 2023; He et al., 2023; Li et al., 2024; Nakhaeinezhadfard et al., 2025), fixing the meta-policy parameters at test time and using context samples as guidance. Many approaches employ a context encoder to map the context to a lower-dimensional *task representation* (Li et al., 2021; Gao et al., 2024; Wang et al., 2023; Zhou

et al., 2024; Zintgraf et al., 2021), though some methods (Xu et al., 2022) use the raw context directly. A fundamental challenge in OMRL is *context distribution shift*: the distribution of context at test time differs from that seen during training. This mismatch can cause the policy to encode task-irrelevant information in the task representation, hindering correct task identification and optimal behavior. Several approaches address this issue. IDAC (Wang et al., 2023) filters context samples based on uncertainty, retaining only high-confidence samples for task inference. CSRO (Gao et al., 2024) applies an information-theoretic framework, minimizing an upper bound on the mutual information between task representations and state-action pairs during training. ER-TRL (Nakhaeinezhadfard et al., 2025) shows that minimizing mutual information between task representations and behavior policies is equivalent to maximizing the entropy of the meta-behavior policy. While most OMRL methods rely on context encoding for task adaptation, (Mitchell et al., 2021) adopts a gradient-based meta-learning approach based on Model-Agnostic Meta-Learning (MAML). MACAW meta-trains the value function using Monte Carlo return estimates and the policy using advantage-weighted regression, and adapts both online through gradient updates using data from test tasks.

## A.2 Adaptation in Reinforcement Learning

Domain adaptation is a core challenge in RL, where the source domain (training environment) differs from the target domain (deployment environment), *e.g.* simulators are used to avoid data collection in the real world. Domain randomization (Zhao et al., 2020; Mehta et al., 2020; Muratore et al., 2022) is heavily used to improve the robustness of the learned policies. This requires modifiable simulators and manually specifying the randomized parameters. System identification methods (Du et al., 2021; Yu et al., 2025; Sobanbabu et al., 2025) estimate the parameters of the simulator from limited real-world data, reducing the sim-to-real gap. A different approach is to train an adaptation module. OSI (Yu et al., 2017) utilizes supervised learning, predicting the dynamics parameters from a short sequence of transitions. Rapid module adaptation (RMA, Kumar et al., 2021; 2022) instead relies on self-supervised learning, predicting the latent space of dynamics parameters from the sequence of transitions. Adaptive control (Cheng et al., 2022; Sung et al., 2024) can compensate for domain shift, using prior knowledge of the dynamical system. All these methods require a simulator/model with changeable parameters. DARC (Eysenbach et al., 2021) trains a classifier and then penalizes the reward for source-domain transitions. Lyu et al. (2024) instead penalizes based on representation mismatch and Xu et al. (2023) filters transitions from a value estimate perspective. SRPO (Xue et al., 2023) regularizes policy optimization in the target domain based on the stationary state distribution; optimal policies in the target and source domains have similar state distributions. ReLCE (Nakhaei et al., 2024) trains a residual policy on top of the source policy to enable adaptation to the target domain.

# B  Experiment Details

We implemented CAMEL with PyTorch (Paszke et al.) and used the Adam optimizer (Kingma & Ba, 2015) for training the models. All neural networks are implemented as MLPs where each intermediate linear layer is followed by Layer Normalization (Ba et al., 2016) and Mish activation function (Misra, 2019).

**Hardware** We used AMD Instinct MI250X GPUs to run our experiments. All experiments have been run on a single GPU with 2 CPU workers and 16GB of RAM.

**Hyperparameters** Table 4 illustrates the hyperparameters for our experiments. We use the same hyperparameters for all of the experiments. We also use the same networks and hyperparameters for a fair comparison, *e.g.* only the context encoder (RNN) and the loss function for training the context encoder are different for DORA and ESCP in Table 1.

**Environments** We now describe the environments and variation factors used for testing. Table 5 provides details of the environments we used, including the dimensionality of the observation and action space, and the distribution of variation factors. Figure 5 visualizes the environments. While the variation factors used in testing are drawn from the same distribution as those in meta-training, we ensure they are distinct from the training samples.

Table 4: Hyperparameters of our method CAMeL.

| Hyperparameter | Value | Description |
|---|---|---|
| **Data Collection** | | |
| Train steps | $10^6$ | |
| Random steps | $5 \times 10^4$ | Num. random steps at start |
| Num. eval episodes | 50 | Num. trajectories in evaluation |
| Eval. every steps | $5 \times 10^4$ | |
| Policy MLP dims | $[512, 512]$ | |
| Value Function MLP dims | $[512, 512]$ | |
| Dropout ratio | 0.1 | |
| Learning rate | $10^{-4}$ | Value function, entropy coeff |
| | $3 \times 10^{-4}$ | Policy |
| Target Entropy | $-\|\mathcal{A}\|_1$ | |
| Batch size | 1024 | |
| Discount factor $\gamma$ | 0.99 | |
| Momentum coef | 0.005 | Soft update target network |
| **Offline Meta-RL** | | |
| Meta batch size | 16 | |
| Batch size | 256 | |
| Context size | 256 | |
| Buffer size | $2 \times 10^5$ | for each task |
| Discount factor $\gamma$ | 0.99 | |
| (Q-)Value Function MLP dims | $[256, 256, 256]$ | |
| Policy MLP dims | $[256, 256]$ | |
| Encoder and Decoder MLP dims | $[256, 256]$ | |
| Task representation dim | 2 | |
| Learning rate | $3 \times 10^{-4}$ | Policy and (Q-)Value functions |
| | $10^{-4}$ | Encoder/Decoder |
| Reconstruction obj weight $\lambda$ | 100.0 | |
| Inverse distance weight $\beta$ | 1.0 | In contrastive obj |
| Expectile regression $\tau$ | 0.7 | |
| Inverse temperature $\mathcal{B}$ | 3.0 | In policy optimization |
| Momentum coef. | 0.005 | soft update target network |
| **Robust BOCPD** | | |
| $\mu_\theta$ | **0** | Prior of mean |
| $\Sigma_\theta$ | **I** | Prior of covariance |
| Scaling Parameter $\alpha$ | 0.1 | Generalized Bayesian Inference |
| Prior Parameter $h$ | 100 | Prior on segment length (uniform) |
| Pruning Parameter $k$ | 20 | Top candidates for computing the posterior |

- **Hopper Friction** and **Walker Friction**: a hopper (one-legged robot) or walker (bi-legged) robot must move as fast as it can while the friction changes to simulate different terrains. The original friction coefficient is scaled by a factor $1.5^u$ where $u \sim [-1.5, 1.5]$.

- **Ant Direction**: an ant (quadruped) robot moving in different desired directions as fast as it can, where the desired direction is sampled from $[-\pi, \pi]$.

- **Cheetah Speed**: a cheetah robot moving forwards/backwards with different desired speeds, where the desired speed is sampled from $[-10, 10]$.

- **Cheetah Length Speed**: a cheetah robot moving forward with different torso lengths and desired speeds. The torso length is sampled from $[0.4, 0.6]$ and the desired speed is sampled from $[3, 8]$.

- **Finger Length Speed**: a planar finger robot with different lengths rotating a body with different desired speeds on an unactuated hinge. The length is sampled from $[0.15, 0.25]$ and the desired speed is sampled from $[5, 10]$.

In the first two environments, the variation factor alters the dynamics. In the next two, it modifies the reward function. In the final two environments, the variation factors influence both the dynamics and the reward function.

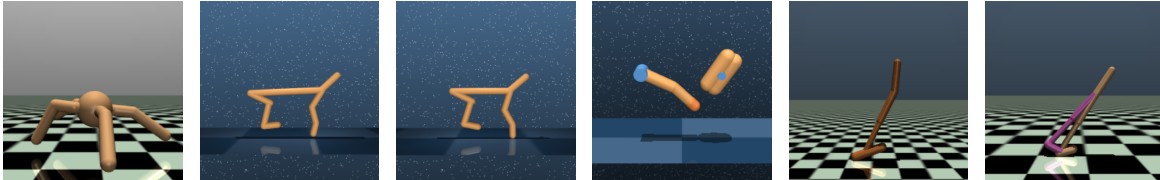

Figure 5: Environments

Table 5: Environment used for evaluation of different methods.

| Environment | Observation Dim | Action Dim | Variation Factors |
|---|---|---|---|
| Ant Direction | 29 | 8 | $\theta \sim [-\pi, \pi]$ |
| Cheetah Speed | 17 | 6 | $v \sim [-10, 10]$ |
| Cheetah Length Speed | 17 | 6 | $v \sim [3, 8] \ \& \ L \sim [0.4, 0.6]$ |
| Finger Length Speed | 9 | 2 | $v \sim [5, 10] \ \& \ L \sim [0.15, 0.25]$ |
| Hopper Friction | 11 | 3 | $u \sim [-1.5, 1.5]$ |
| Walker Friction | 17 | 6 | $u = \{0.8, 1.2\}$ |

## C    Further Results

### C.1    Choice of Offline Meta-RL Algorithm

CAMEL use the OMRL framework to train the context encoder and the policy, and then enable adaptation via change-point detection. In this section, we evaluate our approach with different OMRL methods, where they differ in learning the task representations. Figure 6 summarizes the results for different variants of CAMEL and limiting the context length to $k$ recent transitions for FOCAL and CSRO methods. CAMEL with Different OMRL backbones (FOCAL, CSRO, and UNICORN) outperforms the truncated history baselines, where the robust variant with diffusion score matching has the best performance. Note that we normalize the returns according to the performance of the OMRL agent given the true change points for each OMRL backbone. CAMEL is compatible with different OMRL algorithms as long as they map transitions from different tasks to unique task representations.

### C.2    Computation Cost

Figure 7 compares the computation cost of CAMEL with baselines (time for one step) during evaluation. While CAMEL, particularly with diffusion score matching, is slower than baselines, namely *DORA*, which computes the task representation on a truncated history, it can still be used in real-time control.

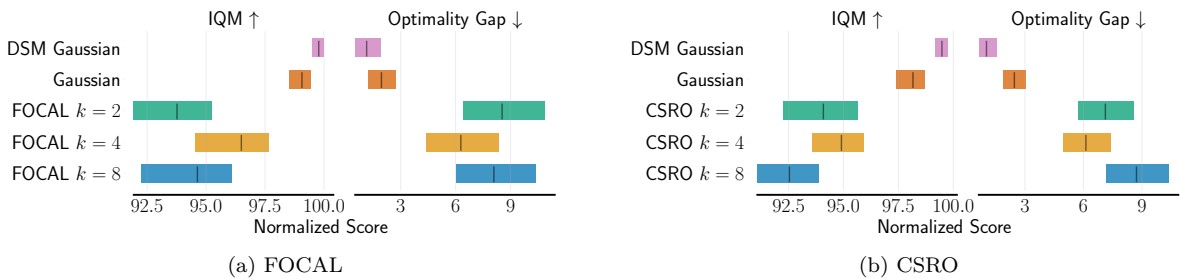

(a) FOCAL                                        (b) CSRO

Figure 6: CAMEL is compatible with different choices of OMRL backbone (FOCAL and CSRO) and outperforms truncating the history of transitions. Aggregate statistics over 6 environments with 5 random seeds.

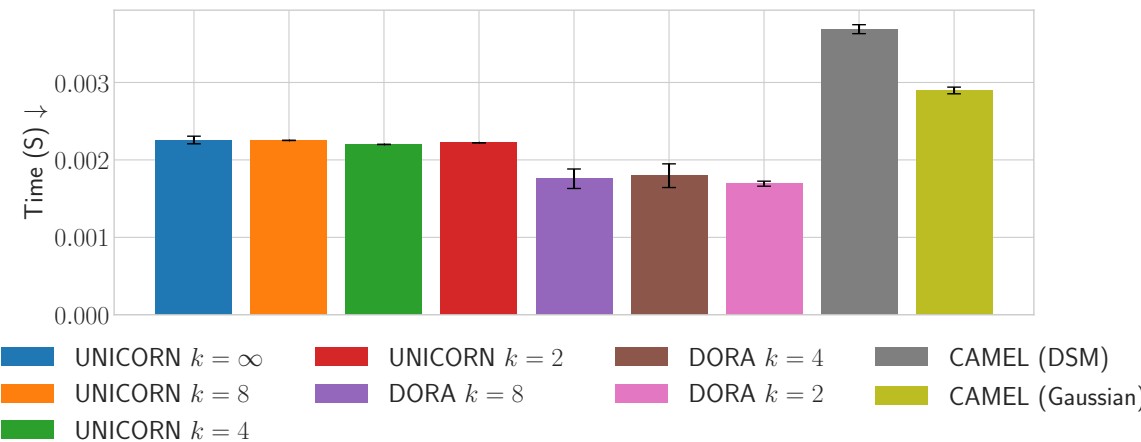

Figure 7: Computation cost of different methods, averaged over 5 random seeds. While CAMeL is computationally more expensive than baselines relying on truncated history, it is still suitable for real-time control.

### C.3 Task Representation Visualization

In Sec. 3.3, we show that diffusion score matching increases robustness to outliers in posterior estimation. We assume that the task representation $\mathbf{z}$ for each task is Gaussian-distributed, and that the context encoder misidentifies the underlying task with probability $\epsilon$. Figure 8 shows the task representations of different transitions for different tasks (indicated by different colors). Task representations corresponding to each task follow different distributions within the same class, which can be approximated by a Gaussian; however, outliers are also present, indicating that the context encoder misidentifies the underlying task for some transitions, underscoring the importance of increasing robustness.

In Sec. 4.2, we investigate how truncated history and change point detection strategies affect the computation of task representation from transitions in testing by visualizing the evolution of task representation $\mathbf{z}_t$. Figures 9 to 13 provides further visualization of the evolution of task representation, illustrating that with truncated history, increasing the number of recent samples $k$ sacrifices adaptation speed for stability in task inference, while CAMeL avoids this trade-off by detecting abrupt changes from the evolution of task representation. Generalized Bayesian inference with (diffusion) score matching improves robustness in detecting changes, leading to better performance in CAMeL.

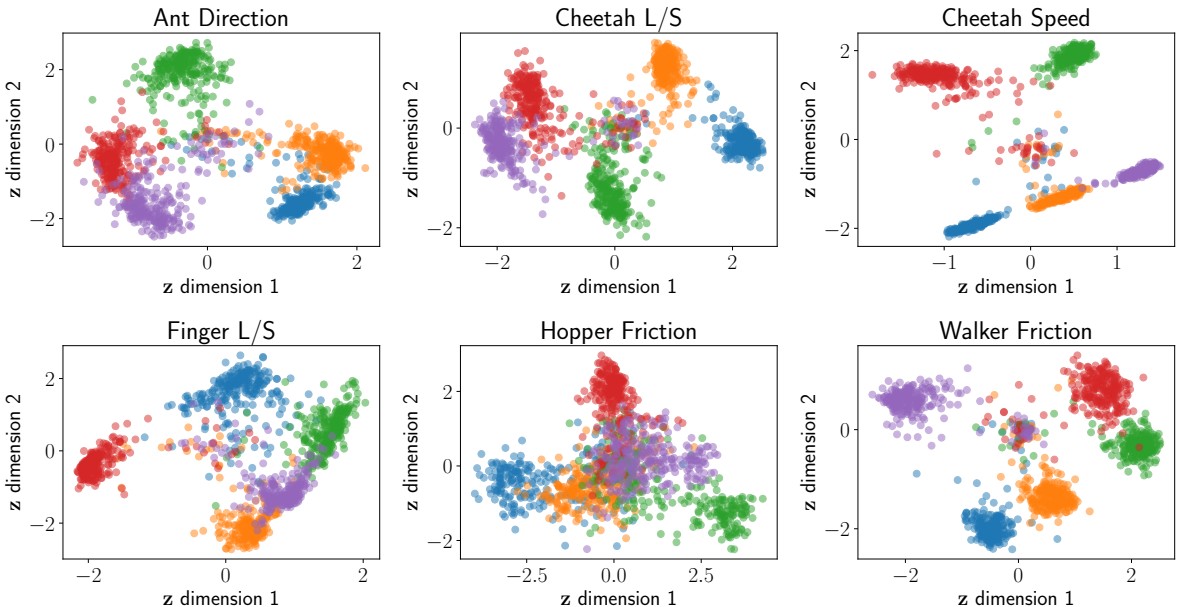

Figure 8: Visualizing task representation for different tasks (different colors). **Context encoder may misidentify the underlying task for some transitions**.

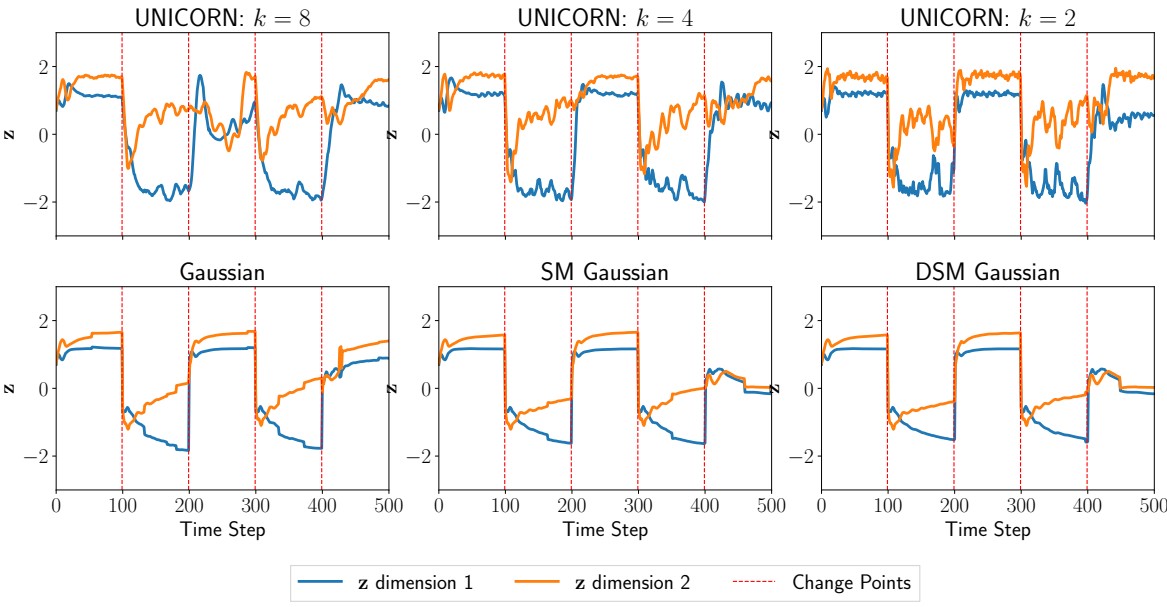

Figure 9: Evolution of the task representation in *Ant Direction*: **Fewer transitions in the truncated history increase adaptation speed at the cost of a noisier task representation. CAMeL avoids this trade-off by detecting the changes online**.

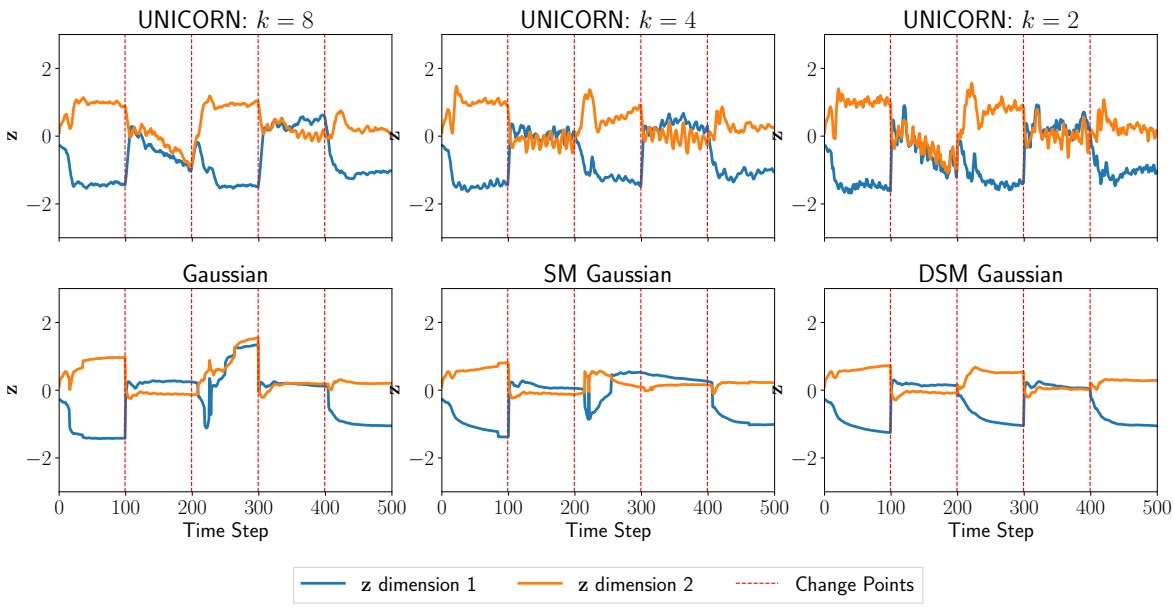

Figure 10: Evolution of the task representation in *Cheetah Speed*

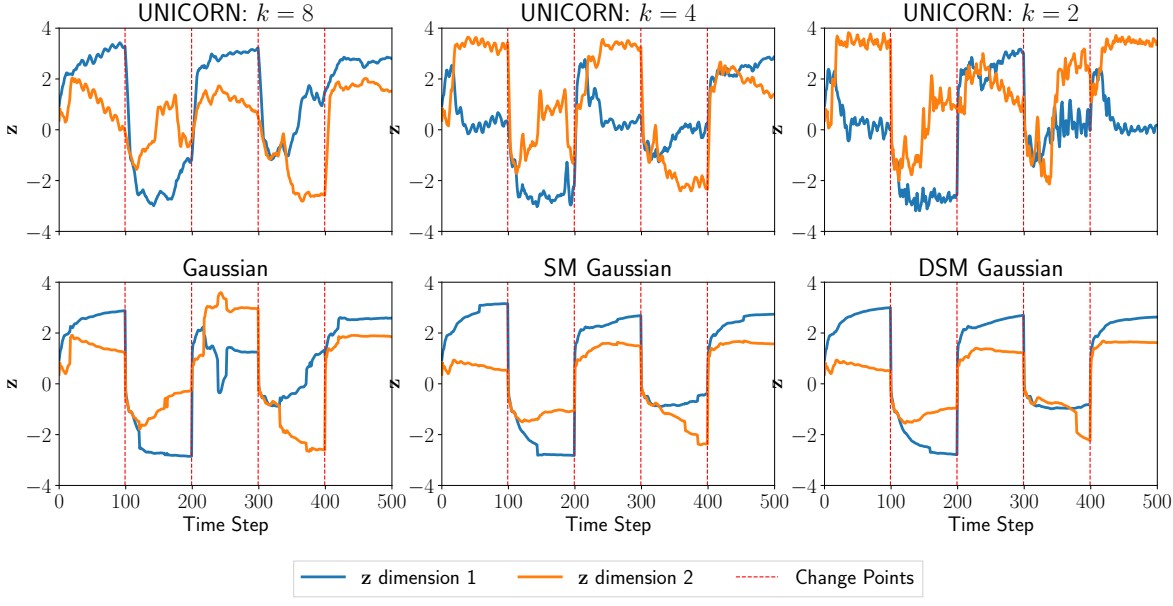

Figure 11: Evolution of the task representation in *Finger L/S*

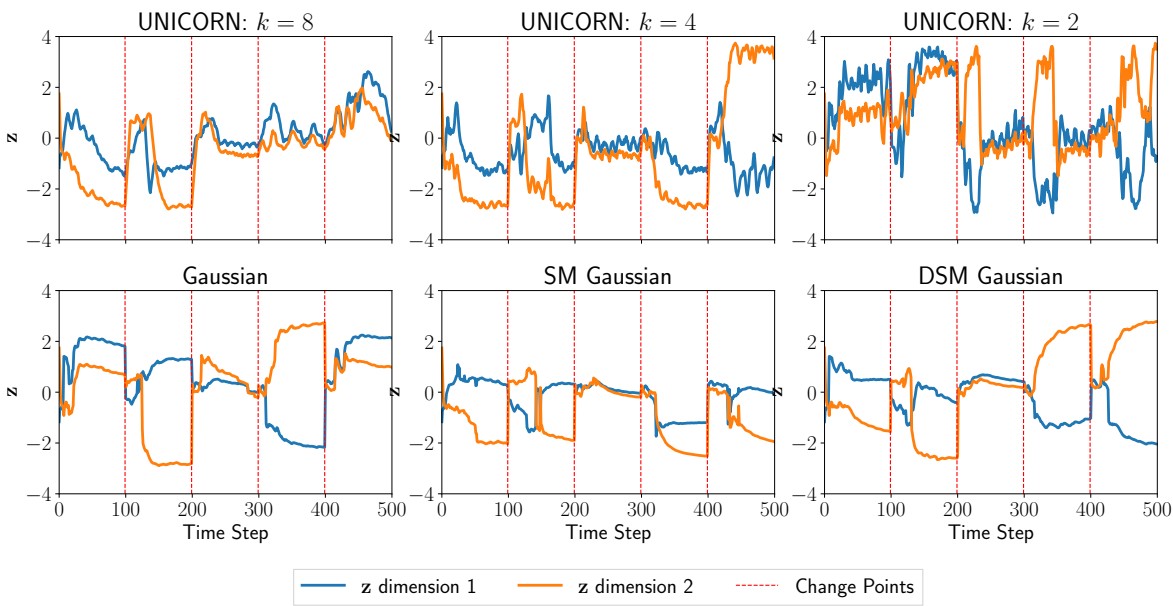

Figure 12: Evolution of the task representation in *Hopper Friction*

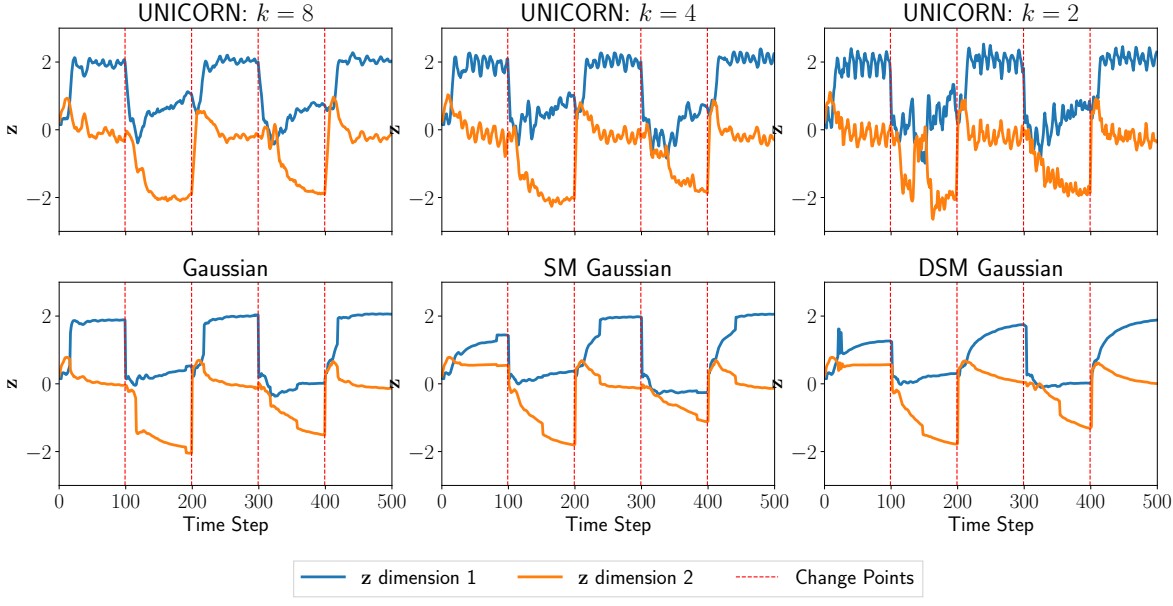

Figure 13: Evolution of the task representation in *Walker Friction*

## C.4    Comparison to Multi-modal Context Encoder

CAMEL utilizes a deterministic context encoder to infer the current task from previous transitions while detecting the abrupt changes from the evolution of task representations. On the other hand, prior work (Bing et al., 2023; Qi et al., 2025) utilizes a multi-modal distribution, *e.g.* Gaussian mixture models (GMMs), for the context encoder to enable adaptation to non-stationary settings. In this section, we investigate a stochastic, multi-modal context encoder in our settings. Inspired by CEMRL Bing et al. (2023), the context encoder outputs a GMM with 3 components and is trained with self-supervised learning with reconstruction of the next state and reward (variational auto-encoding) with KL-regularization. During task inference, the component with the highest average probability over previous transitions is selected, and the task representation $z$ is sampled from a Gaussian distribution corresponding to that component.

Table 6 compares CAMEL (DSM Gaussian variant) to our offline implementation of CEMRL (Bing et al., 2023). For a fair comparison, we use the same offline RL backbone (IQL). CAMEL, which can quickly detect the changes in the environment and adapt accordingly, is more performant than using a multi-modal context encoder.

Figure 14 illustrates the evolution of task representation of our offline implementation of CEMRL. Compared to CAMEL, CEMRL is less sensitive to changes in the environments and adapts more slowly.

Table 6: Comparing multi-modal context encoder for task inference to change point detection in CAMEL. Averaged over 6 random seeds, where ± indicates standard deviations.

| Agent | Ant Direction | Cheetah L/S | Cheetah Speed | Finger L/S |
|---|---|---|---|---|
| CEMRL | $490.4 \pm 55.6$ | $667.5 \pm 50.7$ | $420.6 \pm 46.2$ | $532.6 \pm 38.3$ |
| DSM Gaussian (ours) | $485.6 \pm 20.5$ | $792.9 \pm 1.0$ | $750.8 \pm 8.5$ | $664.6 \pm 5.3$ |

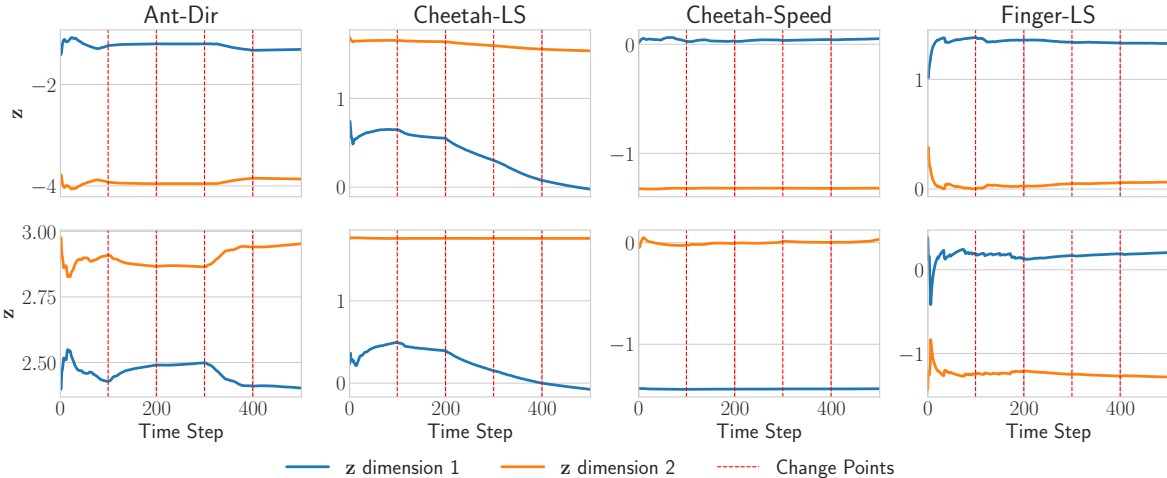

Figure 14: Visualizing the evolution of task representation of CEMRL, where the context encoder outputs a multi-modal distribution (GMM) for task inference. CAMEL detects the changes more quickly.

## C.5    Irregular Abrupt Changes

In this section, we investigate whether CAMEL can adapt to abrupt changes that occur with different intervals. Figure 4 already shows that CAMEL can adapt to irregular changes; however, it only compares the change point detection in different variants of CAMEL without comparing the performance to baselines.

Table 7 compares the performance of different variants of CAMEL to UNICORN with truncated history for task inference. The changes in the environment occur at $[25, 75, 150, 300]$ time steps. We use the

Table 7: Adaptation performance in piecewise stationary environments with irregular change. Average return over 6 random seeds, where ± represents standard deviation. The last three rows indicate variants of CAMeL.

| Agent | Cheetah-LS | Cheetah-Speed | Finger-LS | Hopper-Friction | Walker-Friction |
|---|---|---|---|---|---|
| Oracle | $872.24 \pm 11.19$ | $749.22 \pm 26.61$ | $676.71 \pm 31.36$ | $1331.60 \pm 24.29$ | $1270.09 \pm 70.42$ |
| UNICORN $k = 8$ | $838.63 \pm 9.30$ | $687.76 \pm 36.84$ | $628.85 \pm 31.44$ | $1285.42 \pm 36.02$ | $1206.95 \pm 77.01$ |
| UNICORN $k = 4$ | $836.60 \pm 2.10$ | $729.59 \pm 35.77$ | $623.78 \pm 19.91$ | $1251.10 \pm 46.98$ | $1201.13 \pm 87.74$ |
| UNICORN $k = 2$ | $827.85 \pm 6.65$ | $696.09 \pm 31.10$ | $638.57 \pm 46.16$ | $1221.17 \pm 59.02$ | $1259.70 \pm 46.52$ |
| Gaussian | $848.54 \pm 6.03$ | $730.67 \pm 29.10$ | $653.88 \pm 37.92$ | $1262.91 \pm 31.48$ | $1267.16 \pm 70.47$ |
| SM Gaussian | $861.70 \pm 15.52$ | $740.85 \pm 21.21$ | $660.84 \pm 38.85$ | $1323.90 \pm 27.55$ | $1265.23 \pm 87.10$ |
| DSM Gaussian | $863.95 \pm 10.24$ | $740.11 \pm 15.47$ | $670.39 \pm 34.04$ | $1335.89 \pm 29.46$ | $1263.54 \pm 80.85$ |

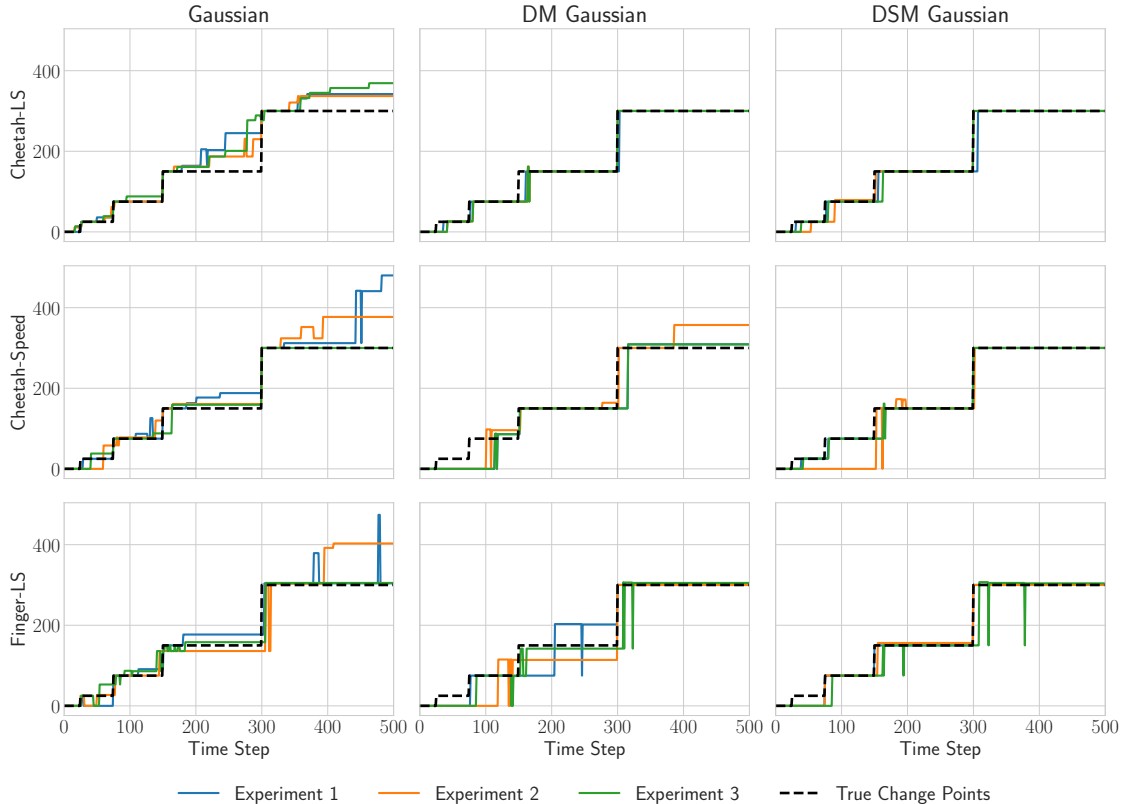

Figure 15: Comparison of change point detection in different variants of CAMeL when changes occur at irregular intervals. DSM Gaussian is more robust to false positives while increasing delays in detection.

same context encoder and offline RL agent for both the truncated history baselines and CAMeL; the only difference is how transitions are selected for computing task representation **z** used as task indicator. Different variants of CAMeL have better performance compared to UNICORN with truncated history. The robust variant (DSM Gaussian) which uses diffusion score matching in generalized Bayesian inference, improves the performance of the naive implementation of BOCPD, consistent with prior results in Table 1.

Figure 15 compares the change point detection in different variants of CAMeL, where changes occur at irregular time steps. Consistent with Figure 4, diffusion score matching with generalized Bayesian inference improves the robustness of BOCPD at the cost of potentially increasing the detection delay, *e.g.*, in the *Cheetah-Speed* environment, the SM Gaussian and DSM Gaussian variants fail to detect some of the changes in some experiments.

## C.6 Ablation: Frequency of Abrupt Changes

In this section, we investigate how the frequency of abrupt changes in the environment affects change point detection and subsequently the performance in testing. Table 8 summarizes the performance for different change intervals $T$, where *Oracle* indicates the agent using only relevant transitions based on privileged information (change points), serving as an upper bound for return. For fair comparison, we use the same agent with the same context encoder in each experiment.

Increasing the frequency of changes (smaller $T$) decreases the return for all agents since accomplishing the underlying tasks and reaching high-reward transitions requires applying suitable actions for several time steps. Also, in certain environments like *Cheetah Speed* where agents need to move forward and backward for different tasks, adaptation can be more challenging than starting from the initial state.

Different variants of CAMEL (Gaussian, DSM Gaussian) outperform truncating history for different change intervals, indicating that detecting changes and using an adaptive-length history consisting of only relevant samples leads to better performance regardless of the frequency of abrupt changes. Diffusion score matching (DSM) improves the adaptation performance of CAMEL in most experiments by improving robustness in detecting change points, as shown in Sec. 4.3 and Sec. C.3.

Table 8: Performance (return) for different change intervals $T$. Detecting changes from the evolution of task representation in CAMEL outperforms truncating the history with different lengths $k$ regardless of the change interval. More frequent abrupt changes reduce performance as the agents have limited time to perform the underlying task. Averaged over 6 random seeds, where $\pm$ indicates 95% confidence interval.

| Environment | Agent | $T = 10$ | $T = 20$ | $T = 25$ | $T = 50$ | $T = 100$ |
|---|---|---|---|---|---|---|
| | Oracle | $189.8 \pm 2.7$ | $229.5 \pm 3.1$ | $250.2 \pm 6.9$ | $442.3 \pm 14.8$ | $494.5 \pm 16.2$ |
| | UNICORN $k = 8$ | $176.3 \pm 3.1$ | $205.4 \pm 5.2$ | $210.2 \pm 4.0$ | $396.5 \pm 4.9$ | $452.0 \pm 18.2$ |
| Ant Direction | UNICORN $k = 4$ | $171.3 \pm 2.1$ | $205.5 \pm 5.4$ | $212.8 \pm 5.8$ | $403.2 \pm 15.5$ | $463.9 \pm 22.9$ |
| | UNICORN $k = 2$ | $176.4 \pm 1.6$ | $207.1 \pm 6.0$ | $210.4 \pm 3.9$ | $401.7 \pm 10.6$ | $455.4 \pm 17.4$ |
| | Gaussian (CAMEL) | $185.3 \pm 1.5$ | $223.7 \pm 4.9$ | $234.3 \pm 6.2$ | $427.4 \pm 7.0$ | $475.2 \pm 17.0$ |
| | DSM Gaussian (CAMEL) | $185.6 \pm 2.4$ | $227.7 \pm 7.2$ | $240.0 \pm 5.8$ | $435.6 \pm 4.9$ | $485.6 \pm 20.5$ |
| | Oracle | $293.5 \pm 0.7$ | $430.1 \pm 0.7$ | $481.4 \pm 0.7$ | $651.5 \pm 1.4$ | $792.1 \pm 10.8$ |
| | UNICORN $k = 8$ | $290.4 \pm 0.8$ | $420.2 \pm 0.7$ | $471.0 \pm 0.7$ | $647.9 \pm 1.7$ | $764.7 \pm 10.1$ |
| Cheetah L/S | UNICORN $k = 4$ | $291.2 \pm 0.7$ | $422.0 \pm 0.6$ | $474.9 \pm 0.6$ | $642.4 \pm 1.3$ | $766.2 \pm 1.7$ |
| | UNICORN $k = 2$ | $291.5 \pm 0.7$ | $425.5 \pm 0.6$ | $480.1 \pm 0.7$ | $645.6 \pm 1.0$ | $764.4 \pm 1.4$ |
| | Gaussian (CAMEL) | $293.5 \pm 0.8$ | $429.6 \pm 0.8$ | $480.4 \pm 0.8$ | $651.6 \pm 1.0$ | $771.6 \pm 10.8$ |
| | DSM Gaussian (CAMEL) | $293.5 \pm 0.8$ | $429.4 \pm 0.8$ | $480.6 \pm 0.6$ | $652.0 \pm 1.3$ | $792.9 \pm 1.0$ |
| | Oracle | $245.2 \pm 1.4$ | $368.6 \pm 5.7$ | $423.4 \pm 9.7$ | $625.2 \pm 7.7$ | $755.2 \pm 7.3$ |
| | UNICORN $k = \infty$ | $230.2 \pm 1.2$ | $337.5 \pm 5.8$ | $380.0 \pm 10.5$ | $579.6 \pm 9.2$ | $707.7 \pm 6.6$ |
| Cheetah Speed | UNICORN $k = 8$ | $242.6 \pm 1.2$ | $351.6 \pm 6.5$ | $408.9 \pm 8.9$ | $613.1 \pm 7.1$ | $733.3 \pm 8.7$ |
| | UNICORN $k = 4$ | $242.7 \pm 1.1$ | $350.3 \pm 5.5$ | $413.1 \pm 9.7$ | $602.3 \pm 9.5$ | $740.0 \pm 8.9$ |
| | UNICORN $k = 2$ | $242.4 \pm 1.3$ | $348.9 \pm 5.2$ | $409.3 \pm 9.1$ | $588.8 \pm 10.5$ | $732.6 \pm 9.0$ |
| | Gaussian (CAMEL) | $244.0 \pm 1.3$ | $363.3 \pm 5.8$ | $419.9 \pm 7.5$ | $614.6 \pm 9.8$ | $746.6 \pm 6.7$ |
| | DSM Gaussian (CAMEL) | $245.0 \pm 1.3$ | $365.7 \pm 4.7$ | $420.0 \pm 9.6$ | $620.0 \pm 11.1$ | $750.8 \pm 8.5$ |
| | Oracle | $428.9 \pm 6.4$ | $545.8 \pm 7.5$ | $556.1 \pm 7.3$ | $623.2 \pm 9.1$ | $670.0 \pm 5.7$ |
| | UNICORN $k = 8$ | $417.1 \pm 8.2$ | $533.6 \pm 8.1$ | $542.0 \pm 7.6$ | $598.4 \pm 8.2$ | $637.5 \pm 9.2$ |
| Finger L/S | UNICORN $k = 4$ | $422.7 \pm 7.3$ | $533.4 \pm 6.9$ | $545.7 \pm 6.1$ | $597.0 \pm 9.8$ | $643.0 \pm 3.5$ |
| | UNICORN $k = 2$ | $420.9 \pm 6.2$ | $530.1 \pm 8.9$ | $548.7 \pm 8.4$ | $598.6 \pm 8.4$ | $641.1 \pm 5.4$ |
| | Gaussian (CAMEL) | $424.4 \pm 7.9$ | $538.3 \pm 7.6$ | $551.0 \pm 8.1$ | $613.0 \pm 8.8$ | $655.9 \pm 7.0$ |
| | DSM Gaussian (CAMEL) | $425.9 \pm 6.8$ | $540.4 \pm 6.3$ | $554.4 \pm 3.8$ | $620.7 \pm 9.7$ | $664.6 \pm 5.3$ |
| | Oracle | $400.3 \pm 3.8$ | $646.0 \pm 5.4$ | $671.2 \pm 5.4$ | $862.6 \pm 0.9$ | $1238.7 \pm 2.0$ |
| | UNICORN $k = 8$ | $395.7 \pm 12.5$ | $610.2 \pm 11.8$ | $656.8 \pm 12.6$ | $845.3 \pm 1.6$ | $1220.7 \pm 1.5$ |
| Hopper Friction | UNICORN $k = 4$ | $394.1 \pm 9.9$ | $629.0 \pm 12.4$ | $652.1 \pm 11.6$ | $845.8 \pm 1.3$ | $1221.0 \pm 5.1$ |
| | UNICORN $k = 2$ | $390.6 \pm 7.6$ | $622.8 \pm 9.4$ | $651.4 \pm 8.5$ | $848.9 \pm 1.8$ | $1219.7 \pm 3.4$ |
| | Gaussian (CAMEL) | $394.2 \pm 6.2$ | $640.0 \pm 10.7$ | $666.8 \pm 10.6$ | $855.4 \pm 1.0$ | $1228.1 \pm 3.0$ |
| | DSM Gaussian (CAMEL) | $393.5 \pm 5.9$ | $641.3 \pm 11.5$ | $664.9 \pm 10.8$ | $856.4 \pm 0.8$ | $1235.4 \pm 2.5$ |
| | Oracle | $303.8 \pm 5.7$ | $725.1 \pm 2.9$ | $740.2 \pm 2.3$ | $838.5 \pm 3.2$ | $1097.2 \pm 3.1$ |
| | UNICORN $k = 8$ | $293.0 \pm 6.7$ | $709.4 \pm 3.2$ | $723.9 \pm 3.1$ | $804.4 \pm 2.3$ | $1057.7 \pm 12.8$ |
| Walker Friction | UNICORN $k = 4$ | $297.9 \pm 6.1$ | $712.0 \pm 3.2$ | $727.9 \pm 3.9$ | $806.7 \pm 2.6$ | $1060.7 \pm 7.6$ |
| | UNICORN $k = 2$ | $295.0 \pm 7.3$ | $717.0 \pm 4.1$ | $729.4 \pm 3.1$ | $801.6 \pm 3.8$ | $1058.7 \pm 9.9$ |
| | Gaussian (CAMEL) | $297.9 \pm 7.2$ | $718.8 \pm 3.2$ | $736.0 \pm 1.4$ | $821.9 \pm 3.0$ | $1065.6 \pm 6.2$ |
| | DSM Gaussian (CAMEL) | $300.4 \pm 7.2$ | $720.8 \pm 3.0$ | $734.7 \pm 2.8$ | $829.1 \pm 2.7$ | $1085.7 \pm 5.7$ |

## C.7 Ablation: Scaling Hyperparameter $\alpha$

In this section, we provide an ablation study on the hyperparameter $\alpha$ used in generalized Bayesian inference with diffusion score matching (Sec. 3.2). Increasing the value of $\alpha$ increases the weight of new samples

in computing $\Sigma_T^{-1}$ according to Equation (13) and subsequently increases the sensitivity of change point detection. Using log-likelihood as the loss function with $\alpha = 1$ leads to standard Bayesian inference.

Figure 16 reports the performance of CAMeL for different values of $\alpha$, and Figure 17 illustrates the change point detection for different values of $\alpha$. When $\alpha$ is too low, change point detection does not detect changes in the environment, and performance suffers as a result. On the other hand, when the value of $\alpha$ is too high, change point detection in CAMeL is too sensitive, and noise in task representations leads to a large number of false detections, hindering performance. However, for a wide range of $\alpha$, CAMeL outperforms truncated history ($k = 4$). Additionally, this optimum range is similar across different environments, enabling the use of the same value for $\alpha$ across different environments. CAMeL uses $\alpha = 0.1$ as the default value.

In *Cheetah Speed*, the optimum range of $\alpha$ is tighter; we hypothesize that since the agent is required to move in different directions at different speeds, errors in change detection have a more significant impact on performance.

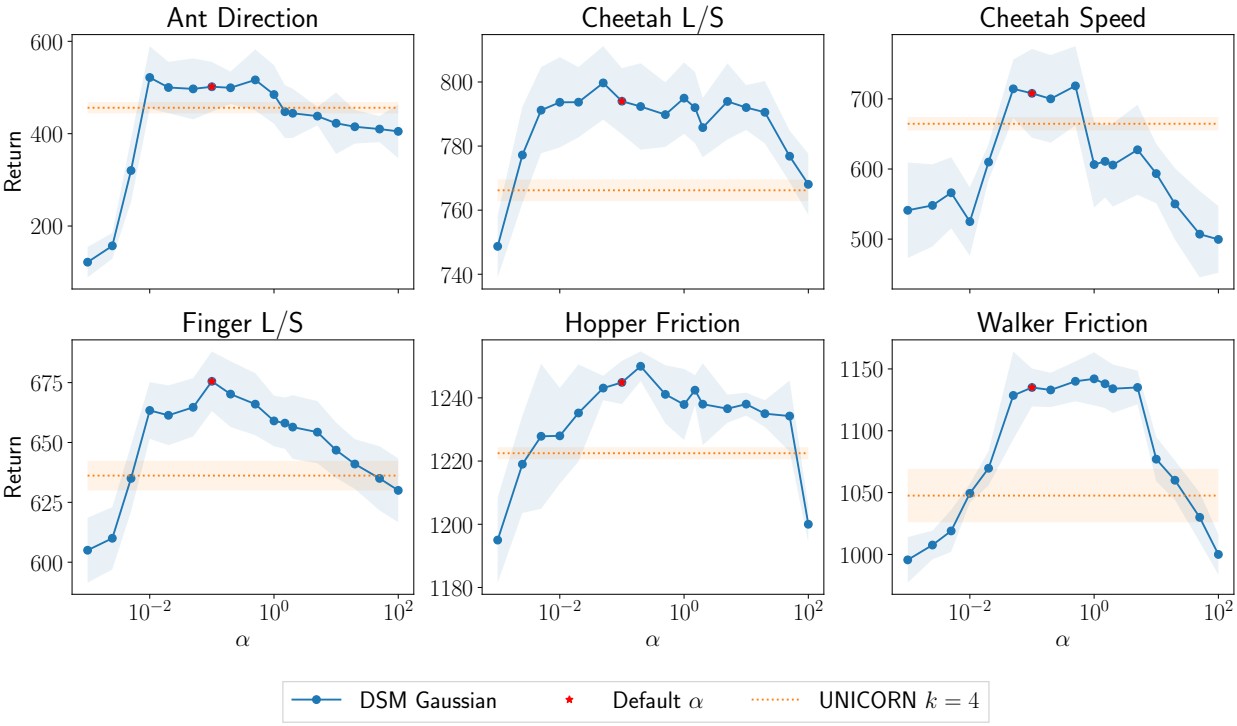

Figure 16: Adaptation performance for different values of $\alpha$, used on generalized Bayesian inference in CAMeL. CAMeL outperforms truncated history for a wide range of $\alpha$, while too small and too high values for $\alpha$ reduce the performance. The shaded area represents a 95% confidence interval over 6 random seeds.

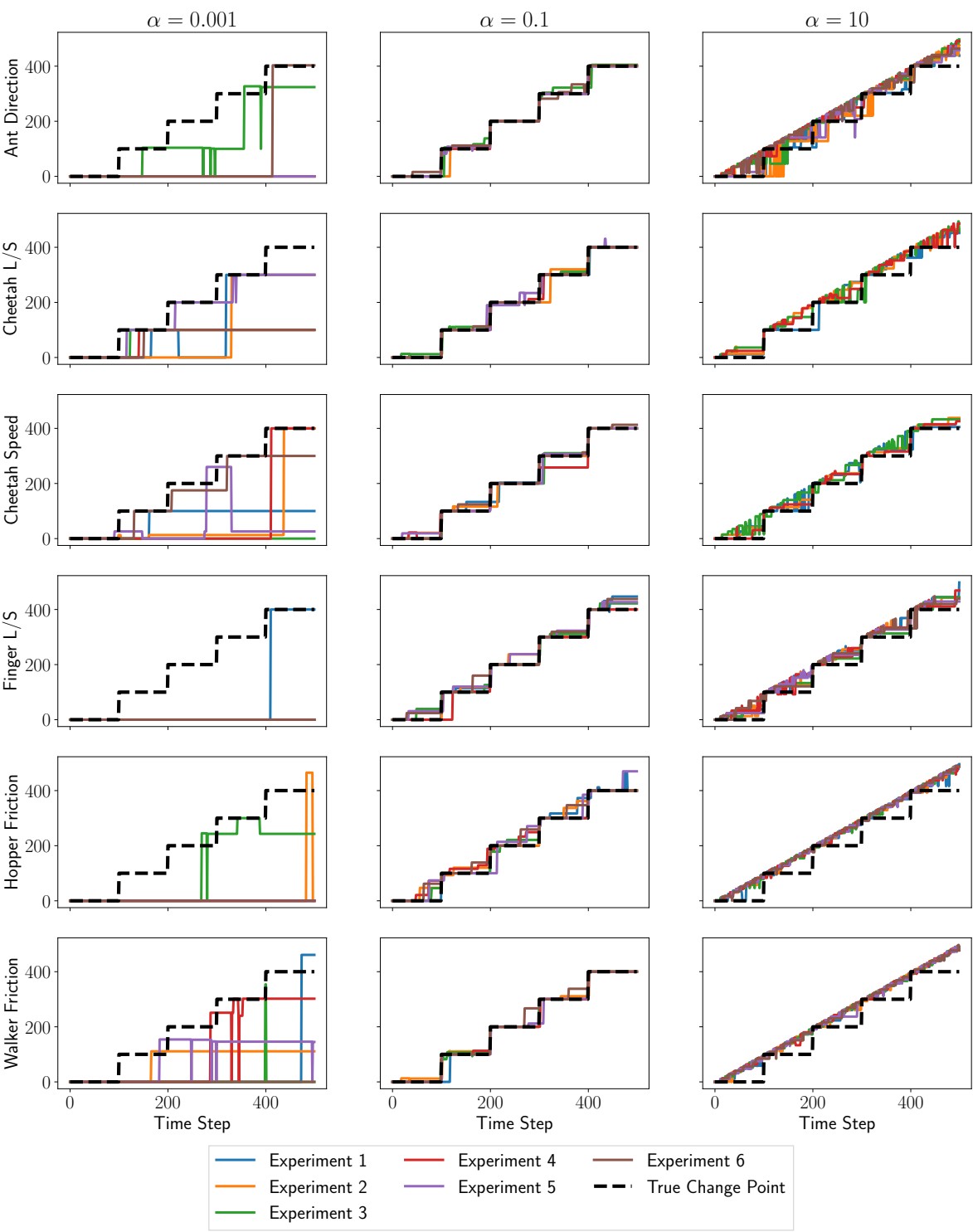

Figure 17: Impact of $\alpha$ on change point detection in CAMEL: Higher values of $\alpha$ increase the sensitivity and lead to false positives, while small values hinder change detection, leading to delays and not detecting.

## C.8 Ablation: Task Representation Dimension

Figure 18 compares the performance and computation cost for different task representation dimensions. Increasing the dimension of $\mathbf{z}$ slightly improves the performance due to better task distinguishability in both CAMEL (DSM Gaussian) and truncating the history (UNICORN $k = 4$), while adding more computation cost to CAMEL than UNICORN $k = 4$. Still, even for $\mathbf{z} \in \mathbb{R}^{15}$, the computation cost of CAMEL does not restrain its applicability.

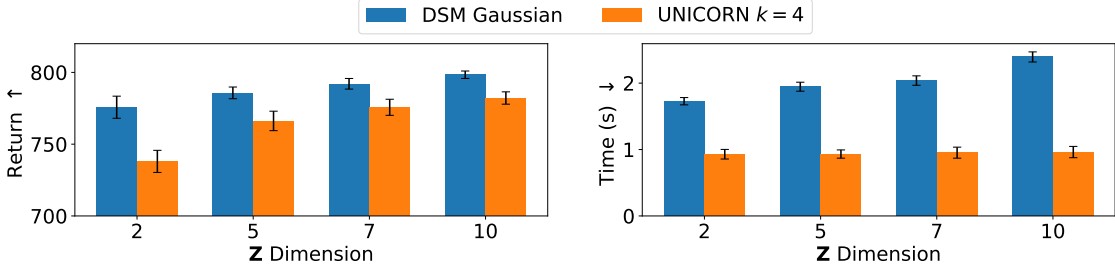

Figure 18: Impact of task representation dimension on performance and computation cost. Averaged over 6 random seeds, and the error bar represents 95% confidence interval.

## C.9 Ablation: CUSUM Change Point Detection

CAMEL utilizes Bayesian online change point detection (BOCPD). In this section, we instead evaluate CAMEL using CUSUM (Cumulative Sum) change point detection. A whitened variable is defined as $\mathbf{x}_t = \Sigma_t^{-1/2}(\mathbf{z}_t - \mu_t)$, where $\mu_t$ and $\Sigma_t$ denote the exponential moving averages (EMAs) of the mean and covariance, respectively. If there is no change in the data stream, the statistical properties (mean and covariance) remain approximately constant, and $\mathbf{x}_t \sim \mathcal{N}(0, I)$. Therefore, $\mathbb{E}[\|\mathbf{x}_t\|_2^2] = d$, where $d$ is the dimension of $\mathbf{z}_t$. Under a change, the mean shifts such that $\mathbf{x}_t \sim \mathcal{N}(\delta, I)$, yielding $\mathbb{E}[\|\mathbf{x}_t\|_2^2] = d + \|\delta\|_2^2$. Therefore, the drift term for change point detection is defined as $s_t = \max(0, s_{t-1} + \|\mathbf{x}_t\|_2^2 - d)$, and a change is detected if $s_t > \delta$. Figure 19 illustrates the performance of CAMEL with CUSUM for different threshold values $\delta$. The accuracy of change point detection, and consequently the overall performance, depends strongly on the threshold $\delta$, where increasing $\delta$ reduces the sensitivity of change point detection. Unlike BOCPD with diffusion score matching, selecting an appropriate value for this hyperparameter is challenging, since the optimal range varies across environments.

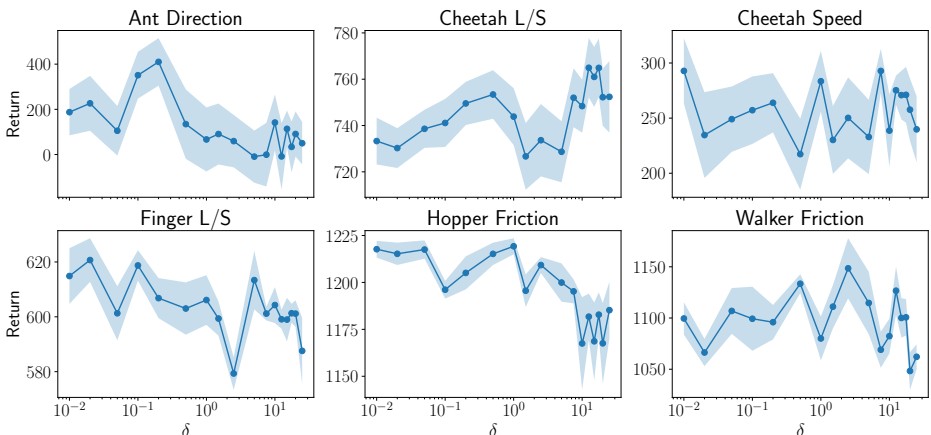

Figure 19: CAMEL with CUSUM change point detection. **The performance is sensitive to the choice threshold** $\delta$ in different environments. The shaded area illustrates 95% confidence interval.

### C.10   Ablation: Prior on Change Point Detection

In this section, we investigate the sensitivity of CAMEL to the choice of prior used in BOCPD. CAMEL assumes a uniform prior for each segment length with $p(l_t \mid l_{t-1}) = \frac{1}{h}$, where $h$ is the uniform distribution parameter with a default value of 100. Figure 20 illustrates the performance for different values of $h$ in different environments. Importantly, the performance is robust to the choice of $h$.

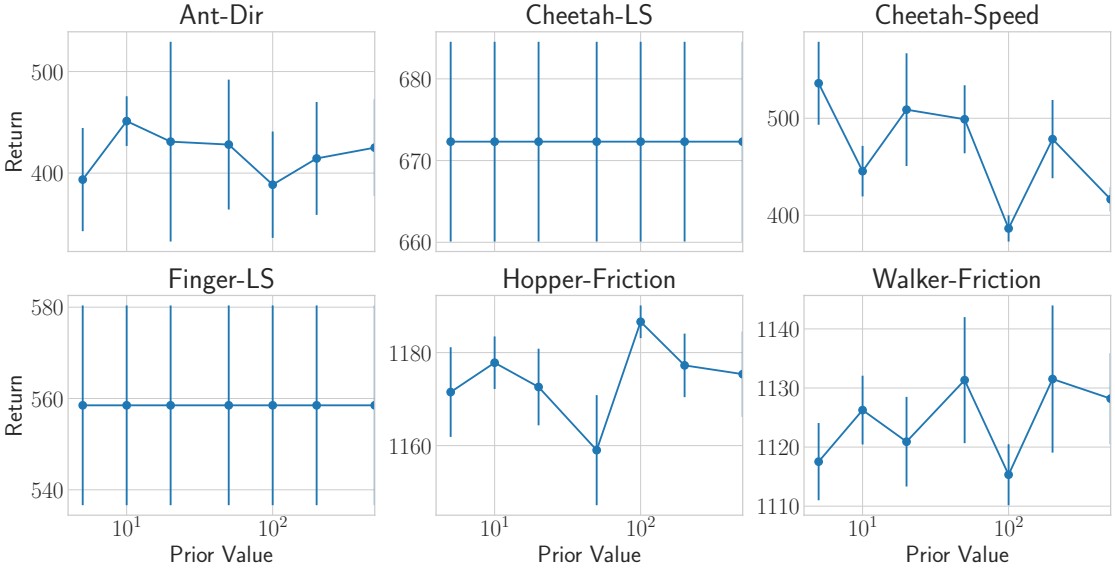

Figure 20: Impact of the prior parameter on the performance. **CAMeL is robust to the choice of prior parameter (uniform distribution)**. Averaged over 6 random seeds, where error bars indicate standard deviation.

### C.11   Ablation: Pruning Parameter

As described in Sec. 3.2 and for computational reasons, CAMEL computes the posterior over the segment length for the top $k$ candidates with a default value of 20. Figure 21 illustrates the impact of this parameter on performance and computation time (one episode) in the *Finger-LS* environment. Increasing $k$ improves the performance and then flattens (around $k = 50$) while increasing the computation complexity.

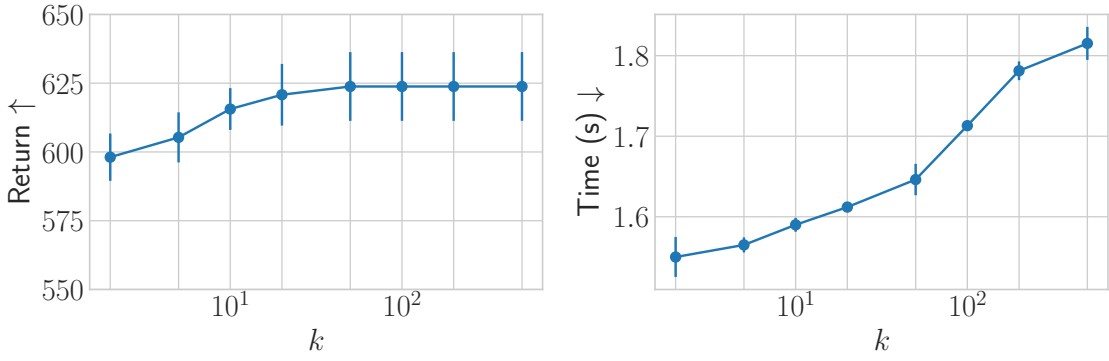

Figure 21: Performance and computation time for different pruning parameter $k$. **Increasing $k$ from low values (2) initially improves the performance while adding computation complexity.** Averaged over 6 random seeds, where error bars indicate standard deviation.

## C.12 Comparison to BOReL and MACAW

In this section, we compare CAMeL with two OMRL methods: BOReL (Dorfman et al., 2021), a context-based approach, and MACAW (Mitchell et al., 2021), a gradient-based meta-learning method. BOReL trains a GRU-based context encoder using reconstruction and KL regularization, together with reward relabeling. MACAW meta-trains the value function and policy, and adapts them to new tasks using online interaction data. For piecewise-stationary evaluation, we update the MACAW parameters after each newly observed transition using a single gradient step.

Table 9 summarizes the results. CAMeL consistently outperforms both BOReL and MACAW under piecewise stationary evaluation, while UNICORN achieves superior performance in the few-shot adaptation setting.

Figure 22 illustrates the evolution of the task representations learned by BOReL across different environments. Compared to BOReL, CAMeL is more responsive to task changes, enabling faster adaptation while maintaining more consistent task representation within each stationary segment.

Table 9: Performance of BOReL and MACAW methods in piecewise stationary and few-shot testing. Averaged over 6 random seeds, where $\pm$ indicates standard deviation.

|  | Agent | Ant Direction | Cheetah L/S | Cheetah Speed | Finger L/S | Walker-Friction |
|---|---|---|---|---|---|---|
| Piecewise-Stationary | BOREL | $302.2 \pm 93.1$ | $263.3 \pm 26.4$ | $206.9 \pm 87.2$ | $385.47 \pm 52.5$ | $748.7 \pm 71.8$ |
|  | MACAW | $212.3 \pm 55.6$ | $250.5 \pm 45.7$ | $220.6 \pm 36.2$ | $320.5 \pm 18.3$ | $670 \pm 86.2$ |
|  | CAMeL | $485.6 \pm 20.5$ | $792.9 \pm 1.0$ | $750.8 \pm 8.5$ | $664.6 \pm 5.3$ | $1085.7 \pm 5.7$ |
| Few-shot | BOREL | $490.4 \pm 55.6$ | $667.5 \pm 5.7$ | $420.6 \pm 16.2$ | $799.1 \pm 78.3$ | $478.3 \pm 50.2$ |
|  | MACAW | $400.2 \pm 36.1$ | $680.5 \pm 4.6$ | $522.1 \pm 7.2$ | $650.6 \pm 68.2$ | $400 \pm 86.8$ |
|  | UNICORN | $685.6 \pm 5.6$ | $886.1 \pm 12.1$ | $717.1 \pm 64.1$ | $950.8 \pm 23.8$ | $541.8 \pm 20.2$ |

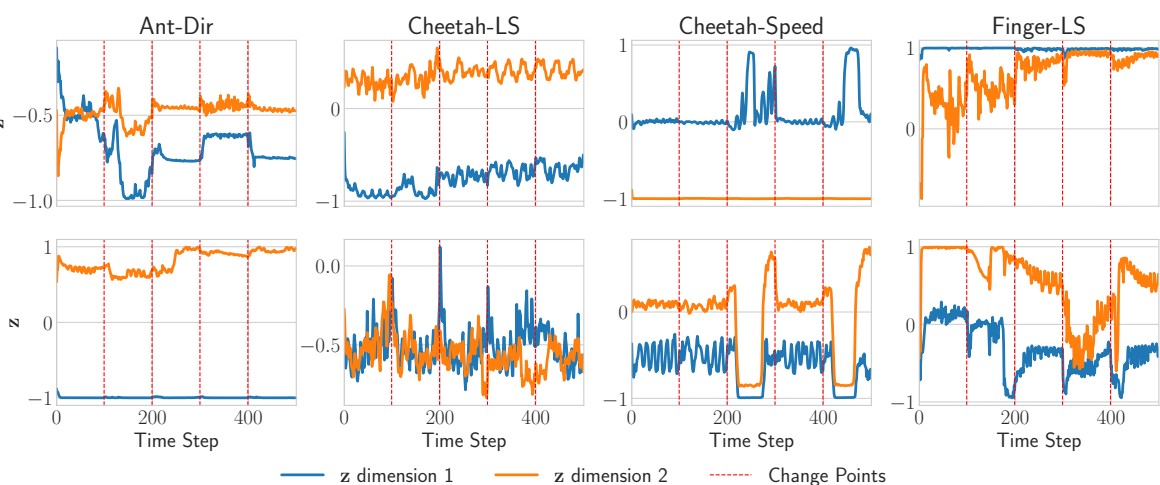

Figure 22: Task representation evolution of BOReL, where each row represents a different random seed.

