# OpenReview forum: "Offline Meta-Reinforcement Learning in Piecewise Stationary Environments"
_TMLR — Under review for TMLR_

### Review · Reviewer_6s4j · 2026-06-22

**Summary Of Contributions:**

The paper introduces Change-Aware Meta Learning (CAMeL), an offline meta-reinforcement learning (OMRL) method tailored for piecewise stationary environments (environments where hidden variation factors remain stable for a period but change abruptly at unknown times). The authors identify a key limitation in existing context-based OMRL methods- relying on a fixed-length context window forces an inherent trade-off between rapid adaptation (that uses a short window) and stable task inference (using a long window).

To overcome this, CAMeL dynamically adjusts its context length by detecting task changes online. The system monitors hidden patterns in the environment to detect exactly when a sudden shift occurs. Because standard change-detection methods often trigger false alarms due to random noise, the authors integrated a specialized filtering technique (diffusion score matching). This makes the system much better at ignoring bad data and only reacting to real changes.


Key Strengths:

1) The problem is well-motivated. The trade-off between adaptation speed and inference stability in fixed-window OMRL is clearly articulated and empirically demonstrated.

2) Combining generalized Bayesian inference with diffusion score matching to robustify change-point detection in an RL context is a clever and effective solution.

3) Comprehensive experiments across several MuJoCo and Contextual DeepMind Control Suite environments show that CAMeL outperforms strong baselines (UNICORN, ESCP, DORA, MetaDT).

4) The paper provides excellent visualizations of latent task representations that illustrates why the method works. Also, the paper includes detailed experiments testing the individual components of the system, breaking down how performance is affected by change intervals, representation dimensions, and hyperparameter scaling.

Key Weaknesses:

1) The method explicitly assumes piecewise stationarity (abrupt, unpredictable changes). It is not clear how gracefully the performance degrades if the environment undergoes gradual/smooth shifts (e.g., slow temperature changes or gradual wear-and-tear).

2) As shown in Appendix C.2 (Figure 7), the robust BOCPD mechanism adds computational cost compared to simple truncated history methods, though the authors note it remains viable for real-time control.

**Additional Comments:**

The paper is well-written and structured. The visual aids specifically the side-by-side plots showing the temporal evolution of the z dimensions under different k constraints versus the CAMeL variants (Figure 3) are commendable. This provides immediate, intuitive clarity to the mathematical arguments presented in Section 3.

**Audience:**

Yes

**Audience Explanation:**

The challenge of non-stationarity is one of the most pressing hurdles for deploying reinforcement learning in real-world, embodied systems like robotics, as simulated in the paper's benchmarks. The TMLR's audience includes researchers in reinforcement learning, meta-learning, and Bayesian methods. This paper successfully applies advanced statistical techniques - diffusion score matching and generalized Bayesian inference to a deep learning pipeline. This intersection of robust statistics and continuous control is highly relevant and could appealing to the community.

**Broader Impact Concerns:**

There are no significant concerns regarding the immediate ethical implications of this work. The paper focuses on foundational algorithmic improvements for adapting control policies in simulated physical environments. While advancements in autonomous control can have dual-use applications (e.g. robotics or automated surveillance), the contributions here are abstract and methodological. A dedicated Broader Impact Statement is not strictly required, though a standard, brief acknowledgment of the dual-use nature of general RL capabilities would be acceptable.

**Claims And Evidence:**

Yes

**Claims Explanation:**

1) Claim 1 (Fixed-history trade-off): Table 1 and Figures 3 and 9-13 consistently show that baselines with smaller k (e.g., UNICORN k=2) adapt faster but suffer from noisy representations, degrading overall return compared to larger k values in stable segments.

2) Claim 2 (CAMeL overcomes this): Table 1 shows CAMeL (both Gaussian and DSM Gaussian variants) achieving the highest normalized returns with statistical significance across almost all tested environments (Ant Direction, Cheetah Speed, Hopper Friction, etc.).

3) Claim 3 (Diffusion score matching reduces false positives): Table 2 explicitly measures detection delay and false positive rates. The "DSM Gaussian" variant achieves a much lower false positive rate (0.2+- 0.4) compared to the naive Gaussian baseline (3.2 +- 1.3), directly supporting the claim that diffusion score matching mitigates the impact of an imperfect context encoder.

**Requested Changes:**

I didn't find any critical adjustments but below are some of the proposed adjustment to strengthen the work-

1) The paper acknowledges in the Limitations (Page 12) that environments can change slowly over time, it would strengthen the paper to include a small empirical evaluation or a more detailed theoretical discussion on how CAMeL behaves when confronted with gradual non-stationarity. Does the change-point detector trigger repeatedly, or does it fail to adapt?

2) Briefly summarize the computational complexity findings (currently in Appendix C.2, Figure 7) within the main text (Section 4). Real-time viability is a major concern for adaptive control and quantifying the inference latency would reassure readers.

3) While Appendix C.5 provides a great ablation on the scaling parameter α, adding a sentence or two in Section 3.2 or 4.1 about how practitioners should go about tuning α for a completely unseen environment would improve reproducibility.

---

> ### Author Response · Authors · 2026-07-22
>
> We thank Reviewer 6s4j for their thoughtful comments. We are pleased that they found CAMEL to be well motivated, clearly presented with informative visualizations, and supported by experiments. Their feedback has helped us further improve the paper.
>
> **RC1: Does the change-point detector trigger repeatedly, or does it fail to adapt?**
>
> We thank the reviewer for raising this important point. Our work focuses on piecewise stationary environments with abrupt task changes. In environments with gradual changes, our initial experiments indicate that CAMEL adapts more slowly, and a truncated-history approach is better suited. However, increasing the scaling parameter $\alpha$ improves the sensitivity of the change-point detector to shifts in the task representation, enabling faster adaptation. However, this is not the a good application of our method CAMEL.
>
> **RC2: Briefly summarize the computational complexity findings**
>
> We thank the reviewer for this suggestion. We have added a brief summary in Section 4.4 (Further Results, main text), highlighting that although CAMEL introduces additional computational overhead, it remains suitable for real-time control, with an average latency of approximately 3.5 ms per control step.
>
> **RC3: Selection of the scaling parameter $\alpha$**
>
> We thank the reviewer for the feedback. We have added a brief discussion in Section 3.2 describing the role of the scaling parameter $\alpha$, its effect on the sensitivity of change-point detection, and the default value used in CAMEL.

---

### Review · Reviewer_2Sgt · 2026-06-26

**Summary Of Contributions:**

This paper studies context-based offline meta-RL in piecewise stationary environments, where hidden variation factors change abruptly at unknown timesteps within an episode. The authors show that fixed-length truncated-context methods face an adaptation-vs-stability trade-off, and propose CAMeL, which runs Bayesian online change point detection (BOCPD) on the stream of task representations and uses only transitions since the last detected change, giving an adaptive context length. To suppress false positives from an imperfect encoder, they use diffusion score matching in generalized Bayesian inference for closed-form, outlier-robust updates. On six non-stationary MuJoCo / DM-Control tasks, CAMeL (especially the DSM variant) nearly matches an oracle with true change points and outperforms truncated-history baselines and MetaDT.

**Strengths**

1. The authors study a well-motivated problem; the context-length trade-off is convincingly demonstrated in both returns and representation visualizations.
2. Detecting changes on the task-representation stream is a clean idea that removes the sensitive context-length hyperparameter.
3. The experiments are comprehensive and code is provided.

**Weaknesses**

1. Novelty is limited. The proposed method is mainly a combination of the robust BOCPD machinery and the OMRL backbone that are well studied in prior work.
2. The detector's assumptions are not examined. Per-transition embeddings are treated as i.i.d. Gaussian despite strong temporal correlation, and its hazard prior encodes an expected segment length equal to the ground-truth change interval in the main experiments.
3. Claimed superiority over multi-modal (GMM) and other change-point-based methods is not backed by any empirical comparison.

**Audience:**

Yes

**Audience Explanation:**

Non-stationary RL, offline meta-RL, and change-point detection are of active interest. The released code adds practical value.

**Broader Impact Concerns:**

No.

**Claims And Evidence:**

Yes

**Claims Explanation:**

The core claims hold up (Table 1, Table 6, Figure 6, Table 2). My main concerns:

1. **Independence assumption.** BOCPD treats per-transition embeddings within a segment as i.i.d. Gaussian, but consecutive transitions are strongly correlated; diffusion score matching helps with outliers, not correlation.
2. **Hazard prior matches ground truth.** The detector assumes an expected segment length of 100 while the main experiments change tasks exactly every 100 steps, contradicting the claim that CAMeL makes no assumption about segment lengths; its sensitivity is not reported.
3. **Missing comparisons.** The comparison is limited to truncated-history methods and MetaDT; the GMM and change-point-based methods cited in Section 5.1 are not evaluated.
4. **Near-oracle normalization.** Scores are normalized so the oracle is 100 and DSM reaches 99+ everywhere, so they measure closeness to a ceiling rather than absolute gains; relatedly, Walker Friction DSM has an unusually large std implying runs above 100.

**Requested Changes:**

Main:

1. Report sensitivity to the hazard prior, and confirm whether it was held fixed in the varying-interval experiments (Table 6).
2. Discuss, and ideally probe, the i.i.d.-within-segment assumption given the temporal correlation between consecutive embeddings.
3. Add at least one empirical comparison against a multi-modal/GMM or change-point-based non-stationary RL method.

Minor:

1. Clarify what is fed to the detector: z is overloaded (per-transition vs. averaged), and the implementation applies an undocumented tanh(·)×5 transform before BOCPD.
2. Discuss scalability beyond the default z-dimension of 2, which matches the at-most-two variation factors in the benchmarks.
3. Clarify the Walker Friction std and how normalized scores exceed 100; include MetaDT in Table 6 / Figure 6 or explain its absence.

---

> ### Author Response · Authors · 2026-07-22
>
> We thank Reviewer 2Sgt for their thoughtful comments. We are pleased that they found CAMEL to be well motivated, clearly presented, and supported by comprehensive experiments. Their feedback has helped us further improve the paper.
>
> **RC1: Report sensitivity to the hazard prior**
>
> We thank the reviewer for this suggestion.
>
> * First, we clarify that CAMEL assumes a **uniform prior** over the segment length, i.e., $p(l_t \mid l_{t-1})=\frac{1}{h}$, where the constant probability mass is determined by the horizon $h$. We have clarified this in Section 4.1.
> * We also include this hyperparameter in Table 4.
> * Finally, we add an ablation study in Section C.10, demonstrating that CAMEL is robust to the choice of $h$.
>
> **RC2: Discuss, and ideally probe, the i.i.d.-within-segment assumption given the temporal correlation between consecutive embeddings**
>
> We thank the reviewer for raising this important point. We agree that task representations are not strictly independent because future transitions depend on the policy, which itself is conditioned on the inferred task representation. However, the learned context encoder reduces this temporal correlation.
>
> We have updated Section 3 to clarify this assumption:
>
> > During training, the context encoder is optimized to map transitions from different tasks to distinct latent task representations, while transitions are sampled randomly from the replay buffer, largely breaking the temporal correlations, similar to off-policy RL. At deployment, CAMEL relies on the learned context encoder to produce task representations with reduced temporal correlation, thereby making the independence assumption more appropriate for change point detection.
>
> **RC3: Empirical comparison against a multi-modal/GMM context encoder**
>
> We appreciate the reviewer's suggestion regarding comparisons with multi-modal context encoders. To address this, we implemented a variant of CEMRL in Section C.4 in which the context encoder outputs a Gaussian mixture model (GMM). Since the official implementation is unavailable, we reproduced the method as faithfully as possible based on the paper. The results show that CAMEL detects task changes more quickly, adapts faster, and achieves higher overall performance.
>
> **RC4: Clarify what is fed to the detector**
>
> We thank the reviewer for this clarification request. The change point detector receives the raw per-transition task representations produced by the context encoder (without temporal averaging). We have revised Section 3 to make this explicit.
>
> **RC5: Discuss scalability beyond the default latent dimension of 2**
>
> In Section C.8 (previously Section C.6), we show that increasing the task representation dimension improves performance at the cost of additional computation. We hypothesize that higher-dimensional latent representations enable the contrastive objective to separate tasks more effectively, resulting in improved performance.
>
> **RC6: Clarify the Walker Friction standard deviation, explain normalized scores exceeding 100, and include MetaDT in Table 6/Figure 6 or explain its absence**
>
> We thank the reviewer for these helpful suggestions.
>
> * We have updated Section 4.1 to explain why normalized returns can occasionally exceed 100. When change points are detected accurately, the inferred task representations produced by CAMEL (DSM Gaussian) can be nearly identical to those obtained using the ground-truth change points. Small differences in the inferred representations, combined with the imperfect conditional policy and the inherent stochasticity of neural network optimization, can occasionally produce slightly higher returns than the expert OMRL agent.
> * Table 8 (previously Table 6) is designed to evaluate how CAMEL handles different task-switching frequencies and compare it against truncated-history baselines. Therefore, we include only UNICORN with truncated histories (using the same trained context encoder and offline RL policy) to isolate the effect of change point detection.
> * Figure 6 is intended to demonstrate that CAMEL is compatible with different offline meta-RL methods (CSRO and FOCAL) and consistently improves over their corresponding truncated-history baselines. MetaDT is not included because it does not rely on a context encoder, making it incompatible with CAMEL's change point detection framework.
> cated history given the same offline RL agent and context encoder.

---

### Review · Reviewer_Q5wx · 2026-07-08

**Summary Of Contributions:**

The paper addresses offline meta-reinforcement learning in piecewise stationary environments, where hidden variation factors (dynamics and/or reward) stay fixed for stochastic periods and then change abruptly at unknown points within an episode. The contributions are:

1. It shows that context-based OMRL using a fixed-length history window faces a trade-off between rapid adaptation (short window, noisy task representation) and stable task inference (long window, slow adaptation), evidenced by a k-sweep over UNICORN/ESCP/DORA (Table 1) and representation visualizations (Fig. 3).
2. It proposes CAMeL, which detects task changes online by running Bayesian Online Change Point Detection (BOCPD) on the stream of task representations produced by the context encoder, then retains only post-change transitions to obtain an adaptive context length (Sec. 3, Alg. 1).
3. It observes that naive BOCPD is prone to false positives under an imperfect encoder, and adopts diffusion-score-matching generalized Bayesian inference (Altamirano et al., 2023) to make detection robust to outliers (Sec. 3.2, Table 2).

On six MuJoCo / DeepMind Control tasks, the diffusion-score-matching variant reaches 99+ normalized return on five of six environments and closely tracks an oracle given the true change points (Table 1), with the result holding across change frequencies (Table 6) and OMRL backbones (Fig. 6).

Strengths: a clean, well-motivated mechanism (detect on the representation stream); strong empirical results against a privileged oracle; thorough ablations isolating the robustness and detection components. Weaknesses: the related work omits core offline meta-RL (MACAW, BOReL) and the closest change-detection prior art (MOLe) and miscasts the cited SeCBAD; a stated detection-delay claim conflicts with Table 2; the dependence on encoder quality is never stress-tested.

**Additional Comments:**

The paper is honest about building on existing components (UNICORN encoder, Altamirano et al. robust BOCPD), which is appropriate for TMLR. The contribution is the composition and the insight to detect on the representation stream, not new detection theory. That insight does survive the prior art above, since MOLe detects on raw experience and model error rather than on a learned task-representation stream; stating the contribution that precisely, against MACAW, BOReL, and MOLe, would sharpen it. Scoping the C3 wording to "we adopt and apply" rather than implying a new derivation would match the evidence and pre-empt novelty objections. The strongest result is that the DSM variant nearly equals the true-change-point oracle; making the encoder-quality dependence and the irregular-interval behavior explicit would let that result carry its full weight.

**Audience:**

Yes

**Audience Explanation:**

The work sits at the intersection of offline meta-RL, non-stationary / piecewise-stationary RL, and Bayesian change-point detection. Each is an active area, and the specific question (adapting context length online under abrupt within-episode change without online exploration) is of direct interest to researchers in offline RL and meta-RL, a clear subset of TMLR's audience. The method is backbone-agnostic (Fig. 6), so the contribution is reusable rather than tied to one architecture, which broadens its relevance.

**Broader Impact Concerns:**

No specific concerns. The work is methodological and evaluated on simulated continuous-control benchmarks with no human data or obvious dual-use beyond generic RL. A dedicated Broader Impact Statement is not required. The existing Limitations paragraph already notes the assumption of abrupt, unpredictable changes; the authors could optionally note the usual sim-to-real caveats for the motivating applications (robotics, resource management), but this is not necessary.

**Claims And Evidence:**

Yes

**Claims Explanation:**

The central claims are supported. The fixed-window trade-off (claim 1) is demonstrated by the k=8/4/2 sweep across three baselines and the representation visualizations. The adaptive-context claim (claim 2) is supported by Table 1 and by Table 6 across change intervals, where CAMeL exceeds every truncated-history baseline and approaches the true-change-point oracle. The robustness claim (claim 3) is supported by the Gaussian/SM/DSM decomposition in Table 1 and the false-positive reduction in Table 2 (3.2 to 0.2 false positives for Gaussian vs DSM). I verified the three load-bearing dependencies (UNICORN, Li et al. 2024; Altamirano et al. 2023; DORA, Zhang et al. 2024), and the submission describes each accurately; the BOCPD machinery is honestly attributed to Altamirano et al. rather than claimed as new.

Two issues qualify the support and should be fixed in revision. First (critical), the Sec. 4.3 claim that robustness costs a delay increase "insignificant (less than one time-step)" is not consistent with Table 2 as written: total delay rises 1.2 to 3.2 from Gaussian to DSM, and SM Gaussian shows 8.0 ± 14.5. The claim needs explicit per-change-point units or softening. Second (critical for accurate positioning), an independent literature search finds core related work absent from the references: MACAW (Mitchell et al., 2021) and BOReL (Dorfman et al., 2021), both offline meta-RL, and MOLe (Nagabandi et al., 2019), which already detects task switches by change-point detection for RL, plus the cited SeCBAD (Chen et al., 2022) miscast in Sec. 5.1. These omissions do not falsify the empirical claims, but the positioning and baseline set are incomplete and must be corrected.

**Requested Changes:**

1. (Critical) Cite and engage the missing related work, and correct the SeCBAD positioning. An independent search finds three core omissions: MACAW (Mitchell et al., 2021) and BOReL (Dorfman et al., 2021), canonical offline meta-RL methods (the first advantage-weighted like CAMeL's own update, the second the offline counterpart of the cited VariBAD), and MOLe (Nagabandi et al., 2019), which already performs online change-point task-switch detection for RL. Cite and discuss all three, and add at least one offline meta-RL baseline (MACAW or BOReL) or justify its exclusion. Separately, restate SeCBAD (Chen et al., 2022) accurately, since it infers a segment-length posterior rather than a GMM (Sec. 5.1), and compare or argue the online/offline gap.

2. (Critical) Reconcile the detection-delay claim with Table 2. Define whether the reported delay is total or per change point, give the per-change-point figures for Gaussian / SM / DSM, and either support or soften "less than one time-step." Address why SM Gaussian incurs the largest delay (8.0 ± 14.5) despite being a robust variant.

3. (Strengthening) Stress-test the dependence on encoder quality. Vary dataset coverage, trajectories per task, or the misidentification rate ε of Assumption 3.1, and report how detection accuracy and return change. This would locate the regime where the method degrades and substantiate the stable-task-identification claim beyond the well-separated cases in Fig. 8.

4. (Strengthening) Report Table 1 under irregular within-trial change intervals. The main table uses changes every 100 steps; the motivation emphasizes unknown change points. A quantitative result with variable segment lengths within a trial would match the stated setting.

5. (Strengthening) Report the pruning parameter k and its effect. State the number of retained segment lengths used in the main experiments (Sec. 3.2) and show detection's sensitivity to it.

6. (Minor) Fix the reference and label issues: the leaked BibTeX fragment in Fearnhead & Liu (2007); the stray characters in the Zhao et al. (2023) title; and "CUMSUM" vs "CUSUM" in Sec. 4.4 and the Sec. C.7 header.

---

> ### Author Response · Authors · 2026-07-22
>
> We thank Reviewer Q5wx for their thoughtful comments. We are pleased that they found CAMEL to be well motivated, clearly presented, and empirically strong. Their feedback has helped us improve the paper.
>
> **RC1: Missing related work, comparison to MACAW and BOReL, and positioning of SeCBAD**
>
> We appreciate the reviewer for identifying the missing related work and citation issues. We have updated the paper as follows:
>
> * We discuss MOLe and SeCBAD in Section 5.1 (Related Work: Piecewise Stationary RL) and clarify how CAMEL differs from these methods.
> * We include MACAW and BOReL in Section A.1 (Extended Related Work: Offline Meta-RL) and compare CAMEL against them in Section C.12. Our results show that while UNICORN achieves stronger few-shot adaptation, CAMEL further improves performance in the piecewise stationary setting.
>
> We thank the reviewer for raising these important points.
>
> **RC2: Reconcile the detection-delay claim**
>
> We acknowledge that the presentation in Table 2 may be misleading. We have added a clarification explaining that the large mean and standard deviation for the *SM Gaussian* variant are caused by a single outlier. This can be observed in Figure 4 (Experiment 1, blue curve), where the method fails to detect one environmental change. Excluding this outlier reduces the detection delay to $2.0 \pm 0.83$. We also clarify that the reported delay is cumulative over the entire episode, as indicated in the table title.
>
> We thank the reviewer for pointing out this ambiguity.
>
> **RC3: Stress-test the dependence on encoder quality**
>
> We agree that CAMEL depends on the quality of the context encoder, which in turn is affected by the coverage and quality of the replay buffer. We have added a discussion of this limitation to the paper.
>
> **RC4: Irregular within-trial change intervals**
>
> We appreciate the reviewer's suggestion regarding irregular change intervals.
>
> * We clarify that CAMEL already detects irregular changes in Sections 4.2 and 4.3 (Figures 3 and 4).
> * We additionally include new experiments in Section C.5 comparing UNICORN with truncated history against CAMEL variants (Table 7) and visualize change-point detection under irregular intervals (Figure 15).
>
> **RC5: Report the pruning parameter $k$ and its effect**
>
> We thank the reviewer for this suggestion.
>
> * We updated Table 4 (Hyperparameters) to include the BOCPD pruning parameter, using the default value $k=20$.
> * We also evaluate its effect in Section C.11. Increasing $k$ from 2 to 50 improves performance before saturating, while incurring additional computational cost.
>
> **RC6: Fix the reference and label issues**
>
> We thank the reviewer for identifying the reference and labeling errors. These have been corrected.

---

### Author Response · Authors · 2026-07-22
**General Response**

We sincerely thank all reviewers for their constructive feedback and positive assessment of our work. We are encouraged that the reviewers found CAMEL to be well motivated, clearly presented, and supported by comprehensive empirical evaluation. Their thoughtful comments have helped us improve both the clarity and completeness of the paper.

In response to the reviewers' suggestions, we have revised the manuscript. **All newly added or modified text is highlighted in violet** to facilitate review. Specifically, we:
- Expanded the related work and clarified the positioning of CAMEL with respect to existing methods, including SeCBAD, MOLe, MACAW, and BOReL, together with additional experimental comparisons where appropriate.
- Added new ablation studies analyzing the sensitivity of CAMEL to hyperparameters, including the hazard prior and BOCPD pruning parameter
- Included additional experiments evaluating CAMEL under irregular task-switching intervals and comparisons against baselines, including a GMM-based context encoder and truncated-history methods.
- Clarified several methodological aspects, including the assumptions underlying the change-point detector, the detector inputs, the cumulative detection-delay metric.
- Discussed important limitations of CAMEL, particularly its dependence on the quality of the learned context encoder and its suitability for piecewise stationary environments with abrupt task changes.
- Corrected reference, labeling, and presentation issues throughout the manuscript.